# DenseMatcher: Learning 3D Semantic Correspondence for Category-Level Manipulation from a Single Demo

**Junzhe Zhu**[*2,5], **Yuanchen Ju**[*3,1], **Junyi Zhang**[4,7], **Muhan Wang**[1]
**Zhecheng Yuan**[1,3,6], **Kaizhe Hu**[1,3,6], **Huazhe Xu**[†1,3,6]

[1]IIIS, Tsinghua University  [2]Tepan Inc.  [3]Shanghai Qi Zhi Institute  [4]UC Berkeley
[5]Stanford University  [6]Shanghai AI Lab  [7]Shanghai Jiao Tong University

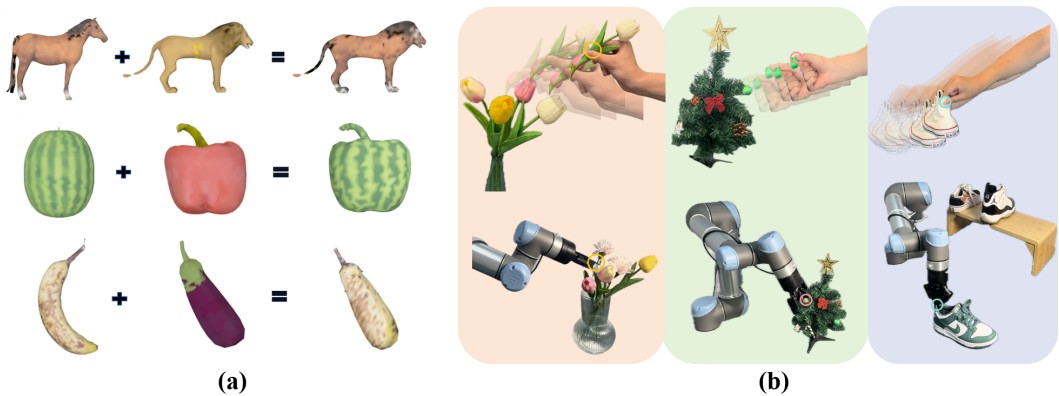

(a)  (b)

Figure 1: (a) **Zero-shot color transfer between 3D assets.** (b) **In real-world robotic experiments**, we use **DenseMatcher** to transfer a manipulation sequence to the robot from a single human demonstration. Circles represent the contact points in the human demo / grasping points for robot manipulation.

## ABSTRACT

Dense 3D correspondence can enhance robotic manipulation by enabling the generalization of spatial, functional, and dynamic information from one object to an unseen counterpart. Compared to shape correspondence, semantic correspondence is more effective in generalizing across different object categories. To this end, we present **DenseMatcher**, a method capable of computing 3D correspondences between in-the-wild objects that share similar structures. DenseMatcher first computes vertex features by projecting multiview 2D features onto meshes and refining them with a 3D network, and subsequently finds dense correspondences with the obtained features using functional map. In addition, we craft the first 3D matching dataset that contains colored object meshes across diverse categories. We demonstrate the downstream effectiveness of DenseMatcher in (i) robotic manipulation, where it achieves **cross-instance** and **cross-category** generalization on long-horizon complex manipulation tasks from observing only **one demo**; (ii) zero-shot color mapping between digital assets, where appearance can be transferred between different objects with relatable geometry. More details and demonstrations can be found at https://tea-lab.github.io/DenseMatcher/.

## 1 INTRODUCTION

Correspondence plays a pivotal role in robotics Wang (2019). By establishing correspondences, we can enable the robot to identify semantically similar components between two objects, which is crucial for various day-to-day manipulation tasks. For instance, in robotic assembly, it is necessary to

---

*Equal contribution,†Corresponding author.

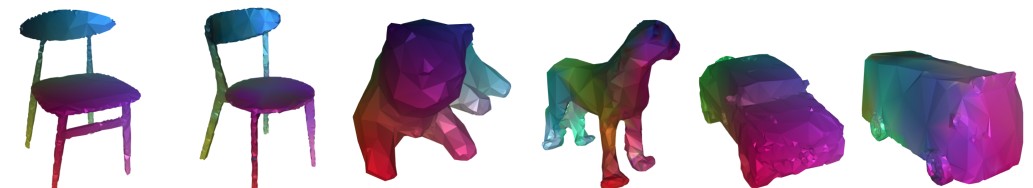

Figure 3: **Predicted correspondences on few-shot categories.** DenseMatcher can generalize across diverse topological variations, given only 5 training examples per category. To ensure that the model is not reliant on canonical spatial poses, we randomly rotate the mesh before the test procedure.

determine the corresponding parts between the target and source objects. Furthermore, recent studies Ju et al. (2024); Kuang et al. (2024) illustrate the capacity to infer the affordances of previously unseen objects through correspondences with a known reference.

Correspondences can be classified along two axes: density and dimensionality. In 2D scenarios, sparse correspondence focuses on matching a limited set of keypoints, while dense correspondence takes spatial proximity into account and aligns every pixel between images. Similarly, in 3D, sparse methods align key points of point clouds or meshes, whereas dense correspondence considers the entire structure for alignment.

Among these types, 3D dense correspondence is particularly advantageous for robotic manipulation, as it ensures continuity by smooth mappings between surfaces. This is crucial for tasks requiring precise multi-point contact and positioning. Additionally, 3D correspondence avoids common 2D ambiguities like distortions from changing viewpoints or occlusions, enhancing a robot's ability to interact accurately with real-world objects.

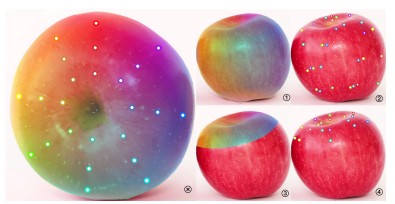

Figure 2: **The 4 types of correspondence.** The reference image is on the left, while the right side demonstrates 1) 3D dense, 2) 3D sparse, 3) 2D dense, and 4) 2D sparse correspondences.

However, existing datasets and methods for 3D dense correspondence (Pratikakis et al., 2016; Dyke et al., 2019; Bogo et al., 2014b; Zuffi et al., 2017; Halimi et al., 2019; Hedlin et al., 2023; Groueix et al., 2018b) often focus on geometry and ignore textures or color information. This limits the ability of models to effectively combine appearance and geometry information, both of which are essential for semantic understanding. In addition, they typically only contain a single or few categories (e.g. humans, four-legged animals), which further limits the generalization ability of models. As a result, prior methods generating dense 3D features can be divided into two categories: (1) 3D networks that only utilize geometry information and are trained on category-specific datasets (Cao et al., 2023; Halimi et al., 2019), which do not generalize well to unseen objects, or (2) models that naively average multiview appearance features from frozen 2D networks (Dutt et al., 2024) and do not utilize any geometry information, which suffer from noise due to varying visibility and pixel coordinates for each vertex, and lack global 3D consistency.

To address this, we release **DenseCorr3D**, the first 3d matching dataset containing colored meshes with dense correspondence annotations, with 600 densely annotated assets across 24 categories. In addition, we develop **DenseMatcher**, a model framework that combines both the powerful generalization capability of 2D foundation models with the geometric undertanding of 3D networks. DenseMatcher first computes per-vertex mesh features by projecting multiview features from 2D foundation models onto 3D meshes, before refining them with a lightweight 3D network. It then calculates dense correspondences using the refined features via functional map, which we improve with several novel constraints.

We further demonstrate the downstream effectiveness of DenseMatcher by performing complex long-horizon **robotic manipulation experiments** based on only a single demonstration of hand-object interaction. Finally, we further showcase the quality of our correspondence by presenting several examples of **color transfer** from one mesh to another without any additional supervision.

In summary, we make the following contributions: (i) a novel 3d matching dataset that remedies the lack of texture information and categories in previous datasets, (ii) a 3D dense correspondence model framework that bridges the gap between 2D and 3D neural networks (iii) comprehensive 3D matching, robotic manipulation, and color transfer experiments.

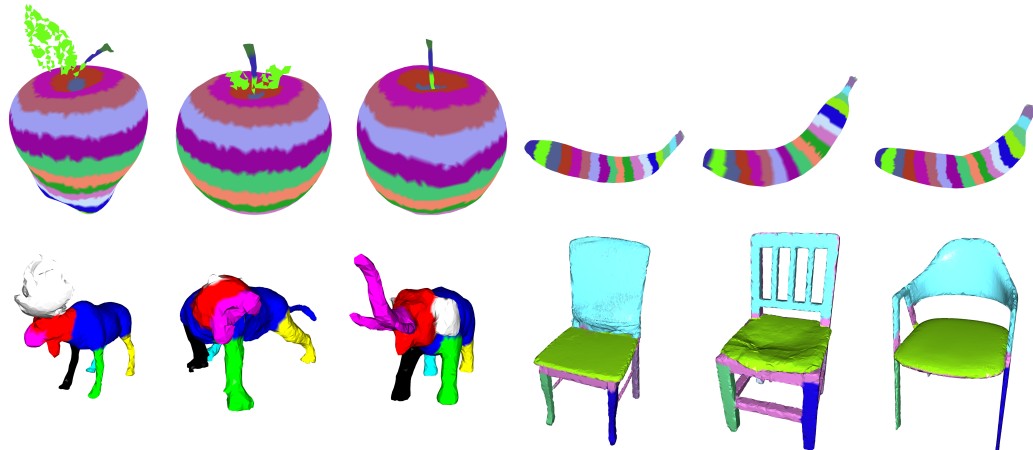

Figure 4: **Semantic group annotations examples** of apple, banana, animals (deer, tiger, elephant), and chairs. Different colors represent different semantic groups across the same category. DenseCorr3D contains objects of varying topologies and structures, both across and within categories.

## 2 RELATED WORKS

**3D Correspondence.** 3D correspondence, or shape correspondence, which focuses on establishing meaningful correspondences between shapes or surfaces, can be categorized into two main approaches: *deformation-based correspondence* and *mapping-based correspondence*. Deformation-based methods focus on tracking an object to its deformed version, often using a canonical template. While simple and direct, these methods do not apply to surfaces under non-isometric transformations (Groueix et al., 2018a;b; Luiten et al., 2024). On the other hand, mapping-based methods such as functional map (Ovsjanikov et al., 2012) establish continuous mappings between two arbitrary surfaces, often leveraging spectral features such as Laplace-Beltrami eigenfunctions (Donati et al., 2020; Roetzer & Bernard, 2024). However, most prior approaches focus on shape features and depend on carefully designed geometric descriptors like Wave Kernel Signature (WKS) (Aubry et al., 2011), or deep features learned from untextured shapes (Cao & Bernard, 2022; Cao et al., 2023), while ignoring semantic relationships between objects.

Recently, powerful 2D foundation models such as DINO (Oquab et al., 2023; Caron et al., 2021) and Stable Diffusion (Rombach et al., 2021) have enabled **deep feature-based 2D semantic correspondence** (Amir et al., 2021; Zhang et al., 2023; Tang et al., 2023; Luo et al., 2024), offering powerful representations extendable to 3D. In particular, Diff3F Dutt et al. (2024) projects such 2D features onto 3D shapes and performs averaging across views. However, Diff3F focuses on untextured shapes and additionally does not incorporate shape information, resulting in noisy and inconsistent 3D features. Our method addresses this by adding a 3D neural network, DiffusionNet (Sharp et al., 2022), to refine 2D features with 3D geometry, producing spatially consistent and informative features.

**Semantic Correspondence for Robotics.** Semantic correspondence helps robots to understand and reason about the relationships within a scene. Florence et al. (2018) utilizes correspondences to map human actions to robots. Recent work Ju et al. (2024) develops a method for few-shot transfer of affordances by querying retrieved objects, and Kuang et al. (2024) extends it to 3D. Xue et al. (2023) infers poses from detected point cloud keypoints for transfering grasps to similar objects with arbitrary poses. Yuan et al. (2024) proposes a multi-view contrastive objective to capture the correspondence under different viewpoints. Notably, leveraging semantic correspondence obviates the need for collecting large amounts of demonstrations (Wang et al., 2024; Ze et al., 2024; Wang et al., 2023; Hu et al., 2024; Chi et al., 2024). Although certain approaches require only a single Liu et al. (2024) or zero demonstrations, they often cannot generalize across diverse object instances and categories.

## 3    TASK: DENSE 3D MATCHING FOR TEXTURED OBJECTS

### 3.1    SEMANTIC GROUPS

To our knowledge, all previous benchmarks Pratikakis et al. (2016); Dyke et al. (2019); Bogo et al. (2014b); Zuffi et al. (2017); Bronstein et al. (2009) on 3d matching focus on category-specific synthetic shapes (e.g. humans, animals) with well-defined vertex-to-vertex correspondences, and lack crucial generalizability to daily objects. None of them includes texture/color information. To remedy this, we develop *semantic groups* as a formal framework for defining category-level 3D correspondence, and release the first 3D matching dataset with textured assets.

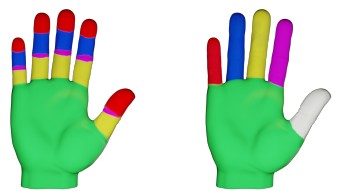

Figure 5: **Two possible partitioning schemes for a hand are shown.**

The definition of correspondence is inherently subjective. For instance, elephant tusks and rhino tusks can correspond based on function, while an elephant's nose can correspond to a rhino's tusk based on location. We formalize this with *semantic groups*. As shown in Figure 4, we group the vertices of each mesh into unique semantic groups, where the group index of $v_i$ is denoted $n(v_i)$ and its group is defined as the set $\mathbb{G}_{v_i} := \{v_j \mid n(v_j) = n(v_i)\}$. Meshes of the same category should share the same semantic groups, with vertices in the same group

having the same learned features, while distinct groups have different features. As illustrated in Figure 5, the partitioning rules for semantic groups can be user-defined. Symmetric objects, like a circular strip around an apple, form single semantic groups, while mirror-symmetric but distinguishable features, like cat ears, may belong to different groups.

## 4    DENSEMATCHER MODEL

### 4.1    PRELIMINARY

**Functional map** (Ovsjanikov et al., 2012) is commonly used for dense 3D correspondences in synthetic meshes but is novel in robotics. We follow the notation from Nogneng & Ovsjanikov (2017) to introduce its formulation. Given source mesh $M$ and target mesh $N$ with $n_M$ and $n_N$ vertices, vertex features $f \in \mathbb{R}^{n_M \times d_{\text{feat}}}$ and $g \in \mathbb{R}^{n_N \times d_{\text{feat}}}$, and diagonal vertex area matrices $A_M \in \mathbb{R}^{n_M \times n_M}$ and $A_N \in \mathbb{R}^{n_N \times n_N}$, we compute the first $k$ eigenfunctions of Laplace-Beltrami operator as a set of spectral bases $\Phi_M \in \mathbb{R}^{n_M \times k}$ and $\Phi_N \in \mathbb{R}^{n_N \times k}$, analogous to sine waves in 1D. Multiplication with these basis or their pseudo-inverses $\Phi^+ = \Phi^T A$ projects functions from the spectral domain to the manifold and back. The map from $M$ to $N$ is represented by a sparse binary matrix $\Pi \in \mathbb{R}^{n_N \times n_M}$, ensuring $g \approx \Pi f$ for corresponding vertices.

Since $\Pi$ is large and takes combinatorial time to solve for, functional map approximates $\Pi$ with a low-rank representation $\Pi = \Phi_N C \Phi_M^+$ where $C \in \mathbb{R}^{k \times k}$ is the *functional map matrix* that we wish to find. We can translate the feature constraint to:

$$g \approx \Pi f = \Phi_N C \Phi_M^+ f \implies \underbrace{\Phi_N^+ g}_{G \in \mathbb{R}^{k \times d_{\text{feat}}}} \approx C \underbrace{\Phi_M^+ f}_{F \in \mathbb{R}^{k \times d_{\text{feat}}}} ,$$

where $G$ and $F$ are low-dimensional projections of $g$ and $f$ onto the eigenfunction basis.

Constraints have been proposed to regularize $C$. Ovsjanikov et al. (2012) shows that if $C$ is isometric, it should commute with the Laplace-Beltrami operator (i.e. left/right multiplying diagonal matrices of eigenvalues $\Lambda_N$ and $\Lambda_M$ with C should be equivalent). To ensure $C$ approximates a point-to-point mapping, Nogneng & Ovsjanikov (2017) enforces that $C$ commutes with point-wise multiplication operators of each feature channel $p$: $X^{(p)} = \Phi_M^+ \text{Diag}(f^{(p)}) \Phi_M$ and $Y^{(p)} = \Phi_N^+ \text{Diag}(g^{(p)}) \Phi_N$.

Combining these constraints with scaling factors $\alpha$ and $\beta$ results in the overall optimization objective:

$$C_{\text{opt}} = \arg\min_C \|CF - G\|_2^2 + \underbrace{\alpha \|\Lambda_N C - C\Lambda_M\|_2^2}_{\substack{\text{isometry constraint:}\\\text{commutativity with}\\\text{Laplace-Beltrami operator}}} + \beta \underbrace{\sum_{p=1}^{d_{\text{feat}}} \|CX^{(p)} - Y^{(p)}C\|_2^2}_{\substack{\text{point-to-point constraint:}\\\text{commutativity with product operator}}}. \quad (1)$$

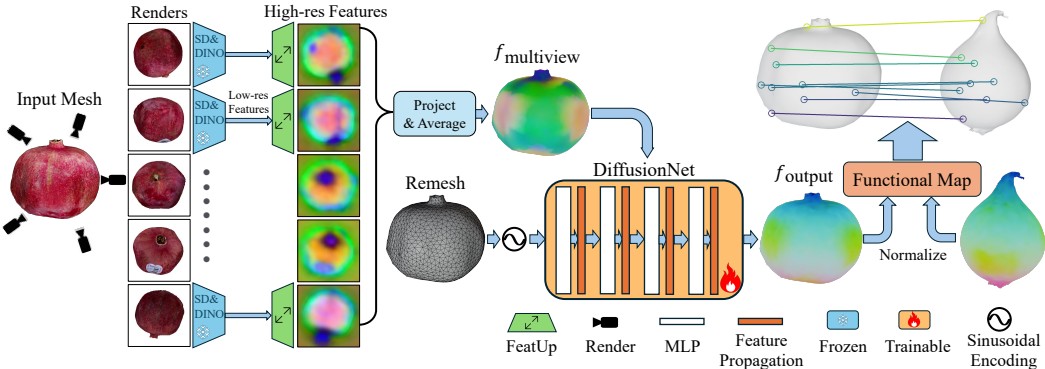

Figure 6: **DenseMatcher model architecture.** SD-DINO (Zhang et al., 2023) fuses 2D features from DINOv2 and Stable Diffusion, which are aggregated and fed into a trainable DiffusionNet. Correspondences are computed from source and target features using functional map.

The detailed derivations can be found in A.4 and Nogneng & Ovsjanikov (2017).

## 4.2 ARCHITECTURE

### 4.2.1 MULTI-VIEW FOUNDATION MODELS (FROZEN 2D "BACKBONE")

We first render multiple views of the 3D asset and compute a 2D feature map for each view using SD-DINO (Zhang et al., 2023), which extracts features with DINOv2 and Stable Diffusion and combines them using the post-processing module of Zhang et al. (2024). We then use Featup (Fu et al., 2024) to upsample the combined feature map. For *each* vertex $v_i$, we retrieve its feature in each view by projecting it into 2D image coordinate and performing bilinear interpolation on the feature map. We then average features from all visible views, or set the feature to a zero vector if the vertex cannot be seen from any view. We dub this aggregated multiview feature $f_{\text{multiview}}(v_i) \in \mathbb{R}^{768}$.

### 4.2.2 DIFFUSIONNET REFINER (TRAINABLE 3D "NECK")

We remesh our 3D asset into ~2000 vertices to obtain its simplified geometry. We incorporate geometry information by concatenating $f_{\text{multiview}}$ with the Heat Kernel Signature descriptor(HKS) (Sun et al., 2009) and sinusoidal positional encoding from Mildenhall et al. (2021) of each vertex's XYZ position. We feed this into DiffusionNet (Sharp et al., 2022), a 3D architecture that alternates between MLP layers and surface feature propagation layers, which serves as the only trainable part of our model. The resulting output is 512-dimensional per-vertex feature $f_{\text{output}}(v_i) \in \mathbb{R}^{512}$, which we then unit-normalize as $f(v_i) := \dfrac{f_{\text{output}}(v_i)}{\|f_{\text{output}}(v_i)\|_2}$.

## 4.3 LOSS FUNCTION

Our loss function consists of two components: $L = L_{\text{semantic}} + L_{\text{preservation}}$. The former ensures that features are similar across nearby semantic groups and distinct across distant groups, while the latter ensures the feature retains rich information learned by 2D foundation models.

### 4.3.1 SEMANTIC DISTANCE LOSS $L_{\text{SEMANTIC}}$

We define the semantic distance between two vertices $D_{\text{semantic}}(v_i, v_j)$ as the average geodesic distance between vertices in their semantic groups. The formal definition can be found in Appendix A.2.1. We design our semantic distance loss to enforce the $L2$ distance between features of any two vertices to scale linearly with their semantic distance. To achieve this, we randomly sample pairs of $v_i, v_j$ on the same mesh and across different meshes, and minimize the negative cosine similarity between $\|f(v_i) - f(v_j)\|$ and $D_{\text{semantic}}(v_i, v_j)$ across sampled pairs of $i, j$:

$$L_{\text{semantic}} = -\cos(\theta) = -\frac{\sum_{i,j} \|f(v_i) - f(v_j)\|_2 \, D_{\text{semantic}}(v_i, v_j)}{\sqrt{\sum_{i,j} \|f(v_i) - f(v_j)\|_2^2} \sqrt{\sum_{i,j} D_{\text{semantic}}(v_i, v_j)^2}}.$$

We prove in A.4.2 that after minimizing this training objective, solving for functional map results in minimal overall $D_{\text{semantic}}(v_{\text{match}(j)}, v_j)$ across all matched pairs.

### 4.3.2 FEATURE PRESERVATION LOSS

We can view our DiffusionNet refiner as an nonlinear operater embedding features from $f_{\text{multiview}}$ into $f_{\text{output}}$. Semantic distance loss ensures the feature space $f(v_i)$ is equipped with a metric that approximates $D_{\text{semantic}}$, but might lose other information learned by 2D foundation models such as object type and material. Therefore, we train a linear layer to approximately invert DiffusionNet and reconstruct $F_{\text{multiview}}$, thereby preserving the rich information learned by SD-DINO:

$$L_{\text{preservation}} = \sum_i^{\|V\|} \|f_{\text{multiview}}(v_i) - W f_{\text{output}}(v_i)\|,$$

where $W \in \mathbb{R}^{768 \times 512}$ is a learnable back-projection matrix that we optimize together with our DiffusionNet parameters.

### 4.4 IMPROVED FUNCTIONAL MAP

After obtaining the vertex features on a pair of meshes, we calculate dense correspondences between them with functional map. Most previous methods using functional map focus on specific shape categories with distinct local geometry such as humans or four-legged animals, and used shape features HKS and WKS. Our approach, however, handles a diverse array of daily objects such as fruits and jugs, which lack distinguishable local features. Despite our learned semantic features, we still observe that the objective of equation 1 is insufficient. In particular, the lack of unique features and large deformations causes $g = \Pi f$ to admit solutions where $\Pi$ is not sparse, leading to noisy correspondences. We therefore propose to add two extra regularization terms:

(1) We clamp the recovered point-to-point mapping matrix $\Pi = \Phi_N C \Phi_M^+$ between $[0, 1]$: $\tilde{\Pi}_{ij} = \max(0, \min(1, \Pi_{ij}))$ and penalize its entropy to promote sparsity:

$$-\sum_{i=1}^{n_N} \sum_{j=1}^{n_M} \tilde{\Pi}_{ij} \log \tilde{\Pi}_{ij},$$

(2) We enforce that each row of $\Pi$ sums to 1 and each column sums to $\frac{n_N}{n_M}$ so that $\Pi$ is a soft assignment matrix:

$$\left( \sum_{i=1}^{n_N} \left( \sum_{j=1}^{n_M} \Pi_{ij} - 1 \right)^2 + \sum_{j=1}^{n_M} \left( \sum_{i=1}^{n_N} \Pi_{ij} - \frac{n_N}{n_M} \right)^2 \right).$$

We scale those to terms and add them to the cost function in equation 1. The detailed optimization procedure can be found in A.2.2.

## 5 THE **DENSECORR3D** DATASET AND BENCHMARK

To remedy the lack of textured data for the 3D matching task, we first filter Objaverse-XL (Deitke et al., 2023) and OmniObject3D (Wu et al., 2023) into 600 instances across 24 categories and split each into train, validation, and test, as specified in Tab.4. For categories with insufficient meshes, we skip the train and validation splits and use those as out-of-distribution training samples.

### 5.1 FILTERING, ANNOTATION AND FORMAT

For fruits and vegetables, we source our meshes from Objaverse-XL. We label landmark points on mesh surfaces, and interpolate them with separate algorithms for each category to acquire ground-truth semantic groups. (See Appendix A.1.2 and A.1.3 for details).

For other daily object categories, we pick assets from OmniObject3D. We use Blender's Vertex Brush functionality to directly label all vertices in each semantic group.

Each instance contains the following: (i) original colored mesh, (ii) remeshed geometry without texture, (iii) ground-truth semantic groups, and (iv) geodesic distance matrix for remeshed geometry

Table 1: **Performance comparison on DenseCorr3D shape matching benchmark.** We report the results on both the full test set and the daily subset. Ablation studies are listed in Section 6.4.

| Methods | All | | Daily | | Fruits & Veges | |
|---|---|---|---|---|---|---|
| | AUC ↑ | Err ↓ | AUC ↑ | Err ↓ | AUC ↑ | Err ↓ |
| ConsistFMap (DenseCorr3D) (Cao & Bernard, 2022) | 0.568 | 6.42 | 0.652 | 5.06 | 0.5404 | 6.87 |
| URSSM (FAUST) (Cao et al., 2023) | 0.485 | 8.19 | 0.602 | 5.90 | 0.445 | 8.96 |
| URSSM (DenseCorr3D) (Cao et al., 2023) | 0.553 | 6.74 | 0.678 | 4.68 | 0.512 | 7.42 |
| Diff3F (Dutt et al., 2024) | 0.437 | 8.20 | 0.562 | 6.25 | 0.396 | 8.85 |
| **DenseMatcher (Ours)** | **0.737** | **3.49** | 0.725 | **3.71** | **0.740** | **3.41** |
| w/o DiffusionNet | 0.708 | 4.02 | 0.726 | 3.75 | 0.702 | 4.11 |
| w/o Preservation Loss | 0.603 | 5.38 | 0.651 | 4.93 | 0.587 | 5.54 |
| w/o Constraint for FMap | 0.706 | 3.84 | **0.729** | 3.72 | 0.698 | 3.88 |

## 5.2 EVALUATION CRITERIA

Given a pair of meshes, we follow the convention of Cao et al. (2023); Cao & Bernard (2022) to compute the Normalized Geodesic Errors (Err) (Kim et al., 2011), and the Area-Under-Curve (AUC) of the threshold-accuracy curve. Since our ground truth annotation is based on semantic groups, we need to update the definition of matching distance to be the distance of the predicted point to the nearest point in the ground truth semantic group. We evaluate correspondence scores across all possible pairs within each category. For example, for categories with 6 test instances, we predict correspondence for $6^2$ pairs of instances.

## 6 EXPERIMENTS

We perform exhaustive evaluation across a spectrum of tasks, encompassing **3D Dense Matching**, **Color Transfer**, and **Zero-Shot Robot Manipulation**. In addition, we perform ablation studies on individual components of our model.

### 6.1 3D DENSE MATCHING

#### 6.1.1 BASELINES

We mainly compare with two training-based deep functional map methods, ConsistFMap (Cao & Bernard, 2022), and URSSM (Cao et al., 2023); and one 2d semantic feature-based method, Diff3F (Dutt et al., 2024). We mainly compare on our proposed DenseCorr3Dbenchmark since our method requires texture as input.

**ConsistFMap (Cao & Bernard, 2022)** utilizes cycle-consistency for robust multi-shape matching across shape collections, making it a strong baseline in unsupervised shape matching. We evaluate its performance when respectively trained on FAUST (Bogo et al., 2014a) and DenseCorr3D.

**URSSM (Cao et al., 2023)** is a state-of-the-art method which extends the functional map framework by coupling point-wise maps and functional maps during learning. We also evaluate both the version trained on FAUST (Bogo et al., 2014a) and on DenseCorr3D.

**Diff3F (Dutt et al., 2024)** projects 2D diffusion-based semantic features onto 3D shapes, focusing on semantic correspondence rather than purely geometric matching. We use the textured mesh as input to the diffusion model to extract the semantic features. For details, refer to A.3.1.

#### 6.1.2 RESULTS

As shown in Tab. 1, we found that our model achieves better AUC and Err compared to the baseline model. Additionally, due to the generalization capability of pre-trained 2D backbones, we achieve much higher accuracy on out-of-distribution test categories listed in Tab. 4 with zero training instances. We also observe surprising qualitative performance on categories with few examples, as shown in Fig. 3.

### 6.2 ZERO-SHOT REAL WORLD ROBOTIC MANIPULATION

We create six real-world manipulation environments, exploring the performance of **DenseMatcher** on daily life tasks by comparing the shape, size, material and category of the manipulated objects.

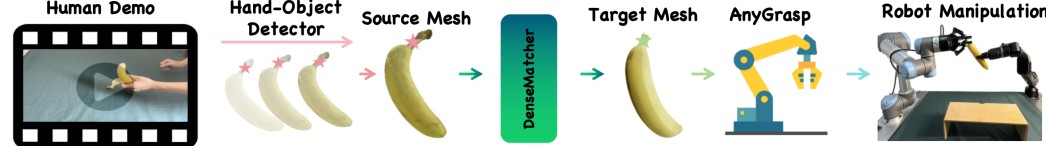

Figure 7: **Real-World Experiment Workflow.** We obtain template mesh and contact points from a human demonstration with hand-object detector (Shan et al., 2020). We then transfer these contact points onto the target mesh with DenseMatcher. Finally, we use off-the-shelf AnyGrasp (Fang et al., 2023) to infer the grasping pose at contact point and proceed with the subsequent manipulation task.

Table 2: **Task difficulty and classification of real world robot manipulation experiments.** The classification of tasks is based on the differences between the objects manipulated in the human demonstration and manipulated by the robot. Here, "cross category" means template and target objects are from different categories. "cross instance" means they are instances of the same category.

| Task Classification | Cross Instance | Cross Viewpoint | Cluttered Multiple Objets | Cross Category | Multiple Keypoints | Long-term | Cross Material |
|---|---|---|---|---|---|---|---|
| Peel a Banana | ✓ | ✓ | | | ✓ | ✓ | |
| Flower Arrangement | ✓ | ✓ | ✓ | | | | |
| Place Shoes | ✓ | ✓ | ✓ | | | ✓ | |
| Decorate Chrismas Tree | ✓ | ✓ | | | | | |
| Pull Out the Carrot | ✓ | ✓ | ✓ | | | ✓ | ✓ |
| Point Object Parts with Pen | ✓ | ✓ | ✓ | ✓ | ✓ | ✓ | |

Table 3: **Real world robot manipulation experiment results.** Robo-ABC[†] (with Original Memory) and Robo-ABC* (with New Memory).

| Task | Peel a Banana | Flower Arrangement | Place Shoes | Decorate Chrismas Tree | Pull Out the Carrot | Point Object Parts with Pen | Overall |
|---|---|---|---|---|---|---|---|
| Robo-ABC[†] | 2/5 | 1/5 | 0/5 | 2/5 | 2/5 | 2/5 | 30% |
| Robo-ABC* | 3/5 | 1/5 | 2/5 | 4/5 | 3/5 | 2/5 | 50% |
| DenseMatcher(Ours) | 4/5 | 3/5 | 4/5 | 5/5 | 4/5 | 3/5 | **76.7%** |

Tasks difficulty and categorization are shown in Tab. 2. We use a RealSense L515 RGB-D camera and a UR5 robot arm to conduct all the real-world experiments. In this section, we use the term *template mesh* to represent the mesh obtained from the human demo, and *target mesh* to refer to the mesh for robot manipulation.

### 6.2.1 GENERAL APPROACH

The workflow of the robotic experiment is shown in Fig. 7.

**Obtaining Human Demonstrations.** After recording RGB-D videos of human demonstrations, we refer to the contact point collection process of VRB (Bahl et al., 2023) and Robo-ABC (Ju et al., 2024) to obtain contact points on the template mesh. For specific details, we recommend referring to the original papers. By using a hand-object detector Shan et al. (2020), we get the contact status between the hand and the object as well as their respective bounding boxes (bbox) in each frame of the video. Then, in the contact frames, we sample the overlapping part of the two bboxes as the contact point. To avoid occlusion, we track the object and trace the contact points back to the first frame, thereby obtaining the template keypoint on the template mesh.

**Contact Point Transfer.** After obtaining the template mesh and keypoints, we calculate the dense descriptors for both the template mesh and the target mesh using DenseMatcher, and find a dense mapping between their vertices using functional map with our proposed improvements. We transfer the keypoints through the dense mapping, thereby obtaining the grasp points on the target mesh.

**Grasp Pose and Post Grasp.** After obtaining the grasp points, we use AnyGrasp to infer the corresponding grasp poses. We provide the waypoints of the trajectory after grasping and the final location to move to after completing the grasp. We use MoveIt! (Coleman et al., 2014) to compute transformation from the target end-effector pose to joint position trajectories.

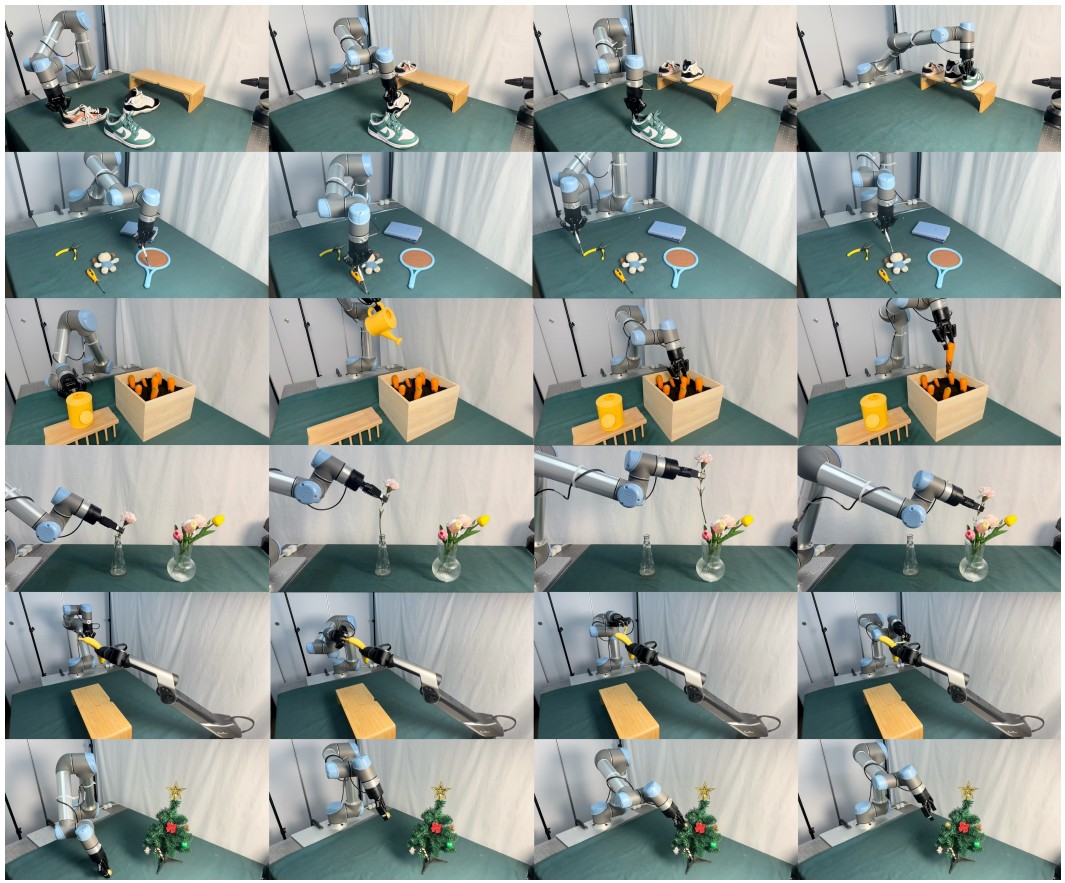

Figure 8: **KeyFrames of 6 robotic tasks.** The tasks from top to bottom are organizing the shoes, pointing object parts with pen, pulling out the carrot, putting flower into a vase, peeling a banana and decorating the the Christmas tree.

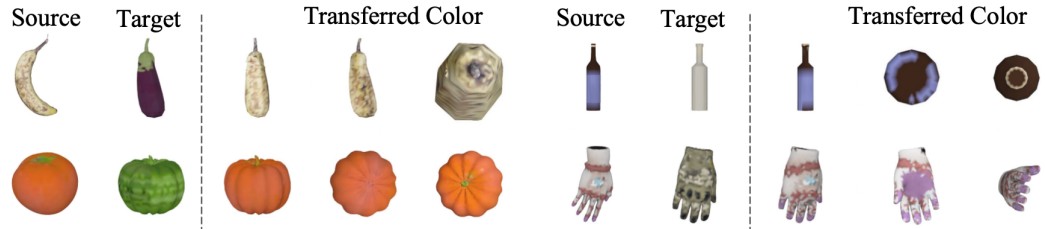

Figure 9: **Color transfer results** between (i) banana and eggplant, (ii) tomato and kabocha squash, and (iii) wine bottles (iii) gloves.

### 6.2.2 BASELINE

Robo-ABC (Ju et al., 2024) utilizes correspondences found in RGB images to transfer affordances. Since Robo-ABC has its own collected affordance memory, we compared two variants: one with full memory capabilities and another where Robo-ABC's affordance memory is only allowed to be collected from the corresponding human demos we provide, while keeping Robo-ABC's original retrieval-and-transfer framework intact.

### 6.2.3 ROBOT MANIPULATION RESULTS

In Tab. 3, we compare the success rate of Robo-ABC with our method in the real world, and use task success rates as the evaluation metric. For each task, we measure the task success rates over five trials. For tasks involving multiple objects and multiple keypoints, we calculate the success rate for each keypoint and object separately and determine the average success rate.

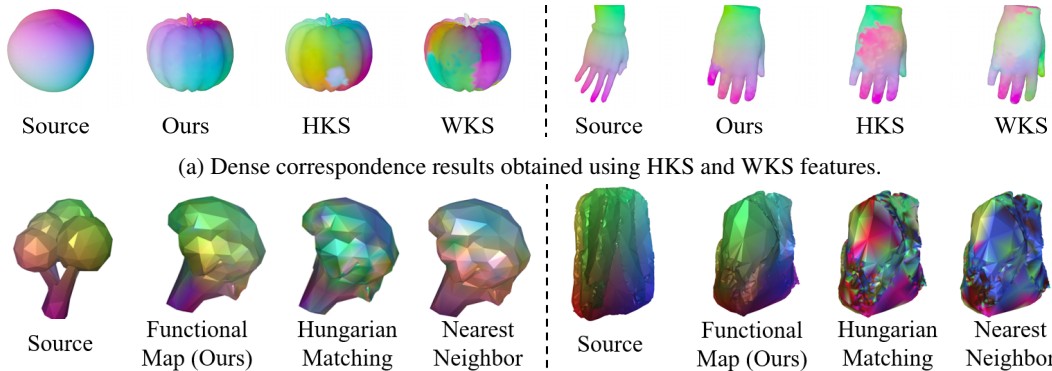

(a) Dense correspondence results obtained using HKS and WKS features.

(b) Dense correspondence results obtained with the same features but different matching methods.

Figure 10: **Ablation study on dense correspondence results.** (a) Effect of using different features (HKS, WKS) with functional maps. (b) Comparison of matching methods using the same features.

### 6.3 COLOR TRANSFER EXPERIMENTS

Ofri-Amar et al. (2023) shows that dense correspondences can be used to transfer object appearances in 2D images. In Fig. 9, we show that our 3D dense correspondence scheme can transfer colors on 3D assets without any additional effort. To our knowledge, this has not been achieved before in the 3D generation literature.

Given a pair of textured meshes and their corresponding simplified meshes, we first color the vertices of the simplified mesh by copying the color from their nearest neighbor on the textured mesh. We then find the point-to-point mapping using DenseMatcher, and directly transfer the color over to the corresponding vertices.

### 6.4 COMPARISON WITH SHAPE DESCRIPTOR FEATURES AND ABLATION STUDIES

As shown in Fig. 10a, we compare functional map outputs using our features to HKS and WKS, shape descriptor features commonly used by prior methods. As can be seen, the mapping obtained with our method significantly outperforms baselines in terms of accuracy and continuity.

As shown in Tab. 1, we perform several ablation studies by (i) skipping DiffusionNet and directly feeding normalized $f_{\text{multiview}}$ into functional map (ii) training our model without loss $L_{\text{preservation}}$, and comparing the difference in evaluation results (iii) removing the proposed entropy penalization and "sum to 1" regularization constraints from the functional map solver.

### 6.5 COMPARISON OF SPATIAL CONSISTENCY

One major advantage of functional map is that it preserves spatial consistency between points, establishing a smooth mapping between surfaces (Cheng et al., 2024). We compare functional map with two baselines in Fig. 10b: Hungarian matching and nearest neighbor retrieval, where we compute a pairwise feature distance matrix between vertices using the same feature from our model. Functional map produces a smooth mapping by preserving both point-wise features and spatial relations between points, while the baselines only preserve the former and result in speckled mismatches.

## 7 CONCLUSION

In summary, we make the following contributions in this paper:
- We create the first 3D matching dataset with colored meshes, containing 600 assets spanning 24 categories with dense correspondence annotation.
- We bridge the gap between 3D and multiview 2D correspondence methods by developing a framework that combines appearance and geometric information by refining features from 2D vision models with 3D geometric models.
- We demonstrate the effectiveness of our approach by performing two sets of experiments that generalize across objects from different categories: (i) real-world robotic manipulation experiments of long-horizon tasks requiring multiple grasps (ii) color transfer experiments.

**Acknowledgments.** We extend our sincere thanks to our colleagues and labmates at the Shanghai Qi Zhi Institute for their contributions to the human demonstrations (in no specific order): **Guangqi Jiang**, **Yuanliang Ju**, **Pu Hua**, **Lesi Chen**, **Huanyu Li** and **Tianming Wei**. We deeply appreciate their beautiful and skillful hands. Special thanks to **Galaxea** for their hardware support of the A1 arm.

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

## A    APPENDIX

### A.1    TASK DESCRIPTION

In this section, we elaborate details of our tasks and showcase our model's generalization capacity by illustrating differences between the objects manipulated by human demonstration and robots.

**Peeling a Banana.** Given an RGBD human demo of peeling a banana, we extract two keypoints: where the banana is initially grasped, and where the base of the banana is held. These points are then mapped to the target banana that the robot will manipulate. Since this task that requires multiple steps with multiple keypoints, we employ a collaborative dual-arm robot approach. We use a Galaxea A1 Arm and a UR5 arm to jointly complete the task of peeling the banana.

**Flower Arrangement.** We provide a human demonstration of arranging flowers. Mimicking human contact points, the robot grasps the flowers by their stem and inserts them into a vase.

**Placing Shoes.** We provide a human demonstration of picking up shoes by the heel and arranging them. The robot operates on three different shoes in a cluttered environment, and and arranges them sequentially throughout multiple steps.

**Decorating the Christmas Tree.** In the human demo for decorating a Christmas tree, the ornaments used are different in shape and color from the objects for robotic manipulation. The robot hangs the ornaments on the Christmas tree to complete the task.

**Pulling Out the Carrot.** The human demonstrator picks up a kettle and manipulates a plush carrot toy. The robot manipulates a real carrot, generalizing across different materials with correspondence. Finally, the robot first picks up the kettle to water the carrot and then pulls out the carrot.

**Pointing Object Parts with Pen.** In this task, we verify the spatial continuity of multiple correspondences on the same object. The template meshes we provide are a ballpoint pen, a plier, a screwdriver, a racket, and a panda toy, while the target objects are different ballpoint pens, pliers, screwdrivers, rackets, and a dog toy. The robot first grasps the ballpoint pen and then uses it to successively poke two keypoints on each object.

### A.1.1    ASSET CATEGORIES

### A.1.2    DATASET FILTERING

For food items, We use Objaverse website's keyword search functionality to narrow our search scope and visually pick meshes that belong to our listed categories. We remove all meshes that are bigger than 300MB in size. For assets from OmniObject3D, we randomly pick meshes from the desired categories. Due to the large sizes of scans, we uniformly sample point clouds from the mesh surface and perform Poisson Reconstruction (Kazhdan et al., 2006) to save downsampled versions of colored meshes for rendering.

### A.1.3    REMESHING

We first normalize object scales by the longest side and multiply them by 0.3, and center the center of bounding boxes at the origin. We cluster each asset into connected components and only keep the largest one. We then remove unreferenced faces and vertices, and merge vertices that are closer than 1/100 of bounding box size. We use the isotropic explicit remeshing filter from PyMeshLab, and iteratively increase the target edge length until the number of vertices goes below the desired vertex number.

### A.1.4    SPARSE KEYPOINT ANNOTATION

For fruits and vegetables with simple geometry, we instruct the annotators to use the 3D annotation tool[1] from StrayRobots to label sparse landmark points, and compute dense vertex groups annotations by interpolation. Specifically, we first manually derive a graph to represent the relationships with keypoints, then, for each pair of connected landmarks, we use the shortest path function of igraph to compute waypoints along certain fractions of the way from one landmark to another. We

---

[1] https://github.com/strayrobots/3d-annotation-tool

| Category | Train | Validation | Test |
|---|---|---|---|
| apple | 65 | 2 | 6 |
| banana | 74 | 2 | 6 |
| bell pepper | 6 | 2 | 6 |
| bread | 52 | 2 | 6 |
| broccoli | 5 | 2 | 6 |
| carrot | 3 | 2 | 6 |
| celery[†] | 0 | 0 | 3 |
| cucumber[†] | 0 | 0 | 6 |
| egg | 17 | 2 | 6 |
| eggplant[†] | 0 | 0 | 3 |
| mushroom | 64 | 2 | 6 |
| onion | 7 | 2 | 6 |
| peach | 13 | 2 | 6 |
| pear | 31 | 2 | 6 |
| potato | 15 | 2 | 6 |
| pumpkin | 40 | 2 | 6 |
| tomato | 12 | 2 | 6 |
| zucchini | 3 | 2 | 6 |
| animals | 5 | 0 | 5 |
| tools | 5 | 0 | 5 |
| vehicles | 5 | 0 | 5 |
| backpacks | 5 | 0 | 5 |
| toiletries | 5 | 0 | 5 |
| chairs | 6 | 0 | 5 |

Table 4: **DenseCorr3D Dataset for each category.** [†] indicates held-out test categories. Animals include: deer, elephant, mouse, cat, giraffe, tiger, panda, leopard, dinosaur. Tools include: kitchen knife, toy knife, pocket knife, hammer, mallet. Vehicles include: car, bus, truck, train head. Toiletries include: shampoo, sun spray, face cream. Chairs include: wood plank chair, velvet chair, veneer chair, mahogany chair, office chair.

then connect different waypoints with the shortest path between them to form dense vertex groups. The average mesh takes 10 seconds to annotate.

### A.1.5 DENSE VERTEX ANNOTATION

For more complex daily object categories, we create a color code chart of semantic groups for each category, and instruct the annotators to use the vertex brush tool from Blender[1] to paint the vertices accordingly. We then use the color codes to parse the painted meshes into separate vertex groups. The average mesh takes 5 minutes to annotate.

### A.2 METHOD DETAILS

### A.2.1 CALCULATION OF SEMANTIC DISTANCE

For two vertices on the same mesh, we perform bipartite matching on the pairwise geodesic distance matrix between vertices in their respective groups and compute the average distance between matched pairs of vertices. If the source and target groups are on different meshes, we find the corresponding group of the source group on the target mesh and compute its distance to the target group analogously. Formally, given vertex $v_i$ and $v_j$, and their semantic groups $\mathbb{G}_{v_i}$ and $\mathbb{G}_{v_j}$ semantic

---

[1]https://www.blender.org

groups with $m$, and $n$ vertices respectively, their semantic distance is defined as:

$$D_{\text{semantic}}(v_i, v_j) = \frac{1}{\min(m,n)} \min_{\substack{\pi_1 \in S_m \\ \pi_2 \in S_n}} \sum_{k=1}^{\min(m,n)} D_{\text{geodesic}}\big(\mathbb{G}_{v_i}(\pi_1(k)), \mathbb{G}_{v_j}(\pi_2(k))\big),$$

$$m = |\mathbb{G}_{v_i}|, n = |\mathbb{G}_{v_j}|$$

where $\pi_1$ and $\pi_2$ are permutation functions encoding bipartite matching.

### A.2.2 FUNCTIONAL MAP SOLVER

We set $\alpha = 10^{-2}$, $\beta = 10^{-4}$, and weigh our added regularization function terms (entropy and sum-to-1) by $10^{-5}$ and $10^{-3}$ respectively. We use the first 10 eigenvectors of the cotangent Laplacian matrix as our bases and zero-initialized C. We modify the implementation from pyFM and compute the gradients of our added terms using Pytorch autograd with CUDA acceleration and use L-BFGS as our solver.

## A.3 EXPERIMENT DETAILS

### A.3.1 DIFF3F BASELINE

The original Diff3F pipeline first renders multiview depth maps, which are used as conditioning to generate colored images with a ControlNet model. It then extracts DINOv2 features from the generated images and combines them with diffusion features to use as multiview features. Since our meshes are textured, we instead directly extract multiview features with Stable Diffusion and DINOv2 from rendered RGB images and concatenate them, before performing dimensionality reduction with PCA. We then aggregate the resulting features onto vertices and feed them into the functional map.

### A.3.2 TRAINING DENSEMATCHER

Our FeatUp module upsamples 16x16 features to 512x512 resolution. We pre-train FeatUp parameters for 10,000 steps on ImageNet (Deng et al., 2009). We freeze the 2D backbone models during training, and optimize a 4-block DiffusionNet with 512 channels on DenseCorr3Dfor 6000 steps with a batch size of 8 using Adam Kingma & Ba (2014). During training, we randomly rotate the meshes, and slice the meshes in half in random directions 50% of the time. We uniformly sample 5 cameras when the meshes are not sliced, and randomly sample 1 or 2 cameras in the same hemisphere when the meshes are sliced. In order to make our model robust to the number of vertices, we randomly set the re-meshing target to between 500 and 2500 vertices during training. In total, training for 50 epochs takes ~12h hours on 8xNvidia A100 GPUs. We do not observe any overfitting in the lightweight DiffusionNet when using a default linear reconstructor, which contains ~5M parameters. Note that in Dutt et al. (2024), 100 views are rendered for each shape, which requires running the computationally heavy 2D extractor 100 times, consuming ~5 minutes for each mesh. Thanks to our 3D network, we found that using only 3 lateral views plus 1 top and 1 bottom view during both training and inferencing is sufficient.

### A.3.3 INFERENCE RUNTIME ANALYSIS

We performed runtime analysis during the inference stage of DenseMatcher on a single A100 GPU. We directly render the original textured meshes to acquire posed images, and found the rendering time to depend on the asset's meshing, varying between ~0.05 seconds to ~3 seconds and averaging to ~0.2 seconds per mesh. Computing 2D SD-DINO features for 5 views each consumes ~3.6 seconds, while performing DiffusionNet forward pass for each mesh consumes ~0.01 seconds. Optimizing the functional map consumes ~0.8 seconds for a pair of meshes with both 500 vertices, and consumes ~2.2 seconds for a pair of meshes with both 2000 vertices. Overall, computing correspondences between a pair of meshes with our algorithm consumes between 8.4 and 12.4 seconds on a single A100 GPU, allowing time-sensitive applications such as robotics planning.

In addition, we ran Hungarian matching on the pairwise vertex feature distance matrix for the 500-vertex case and 2000-vertex case, purely matching features without accounting for spatial consistency. We found the runtime to heavily depend on the sparsity of matrix values. Hungarian matching

takes ~0.01-0.4 seconds for the 500-vertex case, and 0.5-2.5 seconds for the 2000-vertex case. We also derive theoretical runtime from SpiderMatch (Roetzer & Bernard, 2024) and compare them below in Tab. 5.

Table 5: **Runtime of functional map and baselines. All units are in seconds.**

| Method | 500-vertex | 2000-vertex |
|---|---|---|
| Functional Map (our implementation) | 0.8 | 2.2 |
| SpiderMatch (Roetzer & Bernard, 2024) | ~10 | >200 |
| Hungarian Matching (no spatial consistency) | 0.01–0.4 | 0.5–2.5 |

### A.3.4 MODEL PERFORMANCE ON VARYING TOPOLOGIES

Table 6: **3D correspondence performance (Err ↓) on categories with complex topologies.**

| Method | Chairs | Animals | Broccoli | Shampoo |
|---|---|---|---|---|
| URSSM (Cao et al., 2023) | 4.79 | 6.89 | 5.94 | 3.46 |
| **DenseMatcher (Ours)** | **3.74** | **3.29** | **3.18** | **0.73** |

### A.3.5 MODEL PERFORMANCE ON HELD-OUT CATEGORIES

Table 7: **3D correspondence performance on held-out categories.**

| Methods | Celery | | Cucumber | | Eggplant | |
|---|---|---|---|---|---|---|
| | AUC ↑ | Err ↓ | AUC ↑ | Err ↓ | AUC ↑ | Err ↓ |
| ConsistFMap (DenseCorr3D) (Cao & Bernard, 2022) | 0.957 | 0.44 | 0.596 | 4.73 | 0.707 | 3.47 |
| URSSM (FAUST) (Cao et al., 2023) | 0.846 | 1.54 | 0.516 | 5.63 | 0.533 | 6.62 |
| URSSM (DenseCorr3D) (Cao et al., 2023) | 0.926 | 0.75 | 0.601 | 4.62 | 0.621 | 4.64 |
| Diff3F (Dutt et al., 2024) | 0.676 | 3.28 | 0.600 | 4.85 | 0.444 | 7.87 |
| DenseMatcher (Ours) | 0.882 | 1.18 | 0.716 | 3.31 | 0.844 | 1.59 |

## A.4 PROOFS

### A.4.1 PRELIMINARY

We view our source and target mesh as manifold $M$ discretized to $n_M$ vertices, and manifold $N$ discretized to $n_N$ vertices, with diagonal area matrices $A_M \in \mathbb{R}^{n_M \times n_M}$ and $A_N \in \mathbb{R}^{n_N \times n_N}$ denoting the area associated with each vertex. The inner product operator for two scalar functions $x \in \mathbb{R}^n$ and $y \in \mathbb{R}^n$ is defined as:

$$\langle x, y \rangle = x^T A y = \sum_i A_{ii} x_i y_i. \tag{2}$$

Given the area matrix and the contingent weight matrix of the mesh $W \in \mathbb{R}^{n \times n}$ (Meyer et al., 2003), the Laplace-Beltrami operator $\Delta(\cdot)$, which takes a scalar function $x$ on the manifold as input and computes its Laplacian, is defined as:

$$\Delta(x) = (A^{-1} W) x. \tag{3}$$

The first $k$ Laplace-Beltrami eigenfunctions $\Phi_M \in \mathbb{R}^{n_M \times k}, \Phi_N \in \mathbb{R}^{n_N \times k}$ are functions on $M$ and $N$ whose Laplacian is a scaled version of itself, obtained by solving the generalized eigenvalue problem:

$$W\Phi_j = \lambda_j A\Phi_j, \tag{4}$$

where $\Phi_j \in \mathbb{R}^n$ denotes the $j$th eigenfunction and $\lambda_j$ denotes the $j$th eigenvalue. The eigenfunctions are orthonormal w.r.t. the inner produce operator:

$$\langle \Phi_i, \Phi_j \rangle = \begin{cases} 0, & i \neq j \\ 1, & i = j \end{cases}. \tag{5}$$

We compound the $k$ corresponding eigenvalues into diagonal matrices $\Lambda_M \in \mathbb{R}^{k \times k}$ and $\Lambda_N \in \mathbb{R}^{k \times k}$, and re-write the 4 and 5 as:

$$W_M \Phi_M = A_M \Phi_M \Lambda_M \implies \Delta(\Phi_M) = \Phi_M \Lambda_M \tag{6}$$
$$W_N \Phi_N = A_N \Phi_N \Lambda_N \implies \Delta(\Phi_N) = \Phi_N \Lambda_N. \tag{7}$$

Analogous to sine waves which are eigenfunctions of 1d Laplace operator, the Laplace-Beltrami eigenfunctions can serve to project functions back and forth between the manifold and spectral domain.

The pseudo-inverse of the eigenfunction is defined as:

$$\Phi^+ = \Phi^T A. \tag{8}$$

Multiplying a function $x$ with the pseudo-inverse of the bases:

$$X_i = (\Phi^+ x)_i = (\Phi^T A x)_i = \langle \Phi_i, x \rangle, \tag{9}$$

is equivalent to the inner product with the bases, and projects functions from the manifold to the spectral domain, where $X$ is dubbed the "spectral coefficients" of the function, and $X_i \in \mathbb{R}^k$ correspondences to the $i$th eigenfunction.

To obtain a function on the manifold from its spectral coefficients, we can multiply the bases with the coefficients, since:

$$x = Ix \approx \Phi\Phi^+ x = \Phi(\Phi^+ x) = \Phi X. \tag{10}$$

### A.4.2 FEATURE CONSTRAINT IN FUNCTIONAL MAP MINIMIZES SEMANTIC DISTANCE LOSS

We show that the feature matching objective $\|CF - G\|_2^2$ presented in 1 minimizes our proposed semantic distance between matched source and target vertices.

We define our source and target features as $f \in \mathbb{R}^{n_M \times d_{\text{feat}}}$ and $g \in \mathbb{R}^{n_N \times d_{\text{feat}}}$.

We can represent the vertex-to-vertex mapping from $M$ to $N$ with a sparse binary matrix $\Pi \in \mathbb{R}^{n_N \times n_M}$, where:

$$\Pi_{ij} = \begin{cases} 0, & i \neq \text{match}(j) \\ 1, & i = \text{match}(j) \end{cases}. \tag{11}$$

For a pair of corresponding features functions $f \in \mathbb{R}^{n_M \times d_{\text{feat}}}$ and $g \in \mathbb{R}^{n_N \times d_{\text{feat}}}$, we can transport the source feature onto the target mesh with:

$$\hat{g} = \Pi f \implies (\Pi f)_j = \hat{g}_j = f_{\text{match}(j)}, \tag{12}$$

where $\hat{g}$ is the feature for $j$the vertex on target mesh, obtained from its corresponding $\text{match}(j)$th vertex on the source mesh.

From our training objective in Eq 4.3.1, the feature distance should be linearly proportional to the semantic distance function, assuming our training objective is fully optimized. We denote this linear constant as $s$:

$$\|f_i - g_j\|_2 \propto D_{\text{semantic}}(v_i, v_j) \implies \|f_i - g_j\|_2 = sD_{\text{semantic}}(v_i, v_j). \tag{13}$$

Therefore, minimizing the sum of L2 distance between the transported source feature and the target feature is equivalent to minimizing the total semantic distance between matches:

$$\|\Pi f - g\|_2 = \sum_j \|(\Pi f)_j - g_j\|_2 \tag{14}$$

$$= \sum_j \|\hat{g}_j - g_j\|_2 = \|f_{\text{match}(j)} - g_j\|_2 \tag{15}$$

$$= s \sum D_{\text{semantic}}(v_{\text{match}(j)}, v_j). \tag{16}$$

Provided with the first $k$ Laplace-Beltrami eigenvectors, we can decompose $\Pi$ into its low-rank "functional map matrix" representation $C \in \mathbb{R}^{k \times k}$:

$$\Pi = \Phi_N C \Phi_M^+. \tag{17}$$

Therefore,

$$\frac{1}{s} \sum D_{\text{semantic}}(v_{\text{match}_j}, v_j) = \|\Pi f - g\|_2$$

$$= \|\Phi_N C \Phi_M^+ f - g\|_2$$

$$\approx \|\Phi_N C \Phi_M^+ f - \Phi_N \Phi_N^+ g\|_2$$

$$= \|C \Phi_M^+ f - \Phi_N^+ g\|_2$$

$$\leq \|\Phi_N\|_2 \|CF - G\|_2$$

$$\text{where} \quad F := \Phi_M^+ f \in \mathbb{R}^{k \times d_{\text{feat}}}, \quad G := \Phi_N^+ g \in \mathbb{R}^{k \times d_{\text{feat}}}$$

when using the full-rank bases, the inequality on the third line will become equality.

Thus, we prove that minimizing the first term $\|CF - G\|_2^2$ in equation 1 is equivalent to minimizing overall $D$semantic distance between matches in source and target match.

### A.4.3 COMMUTATIVITY WITH LAPLACE OPERATOR INDUCES ISOMETRIC MAPPING

**Lemma A.1.** *For a function $x \in \mathbb{R}^n$, its Laplacian $\Delta(x)$ can be computed as $\Phi \Lambda X$ from its full-rank spectral coefficients $X = \Phi^+ x \in \mathbb{R}^k$.*

**Proof**:

$$\Delta(x) = \Delta(\Phi X) \tag{18}$$

$$= \Delta(\sum_i \Phi_i X_i) \tag{19}$$

$$= \sum_i X_i \Delta(\Phi_i) \tag{20}$$

$$= \sum_i X_i \Phi_i \lambda_i \tag{21}$$

$$= \Phi \Lambda X \tag{22}$$

$$= \Phi \Lambda \Phi^+ x \tag{23}$$

The third line follows from the linearity of the Laplace-Beltrami operator.

On Riemannian manifolds, a diffeomorphism is isometric if and only if the Laplace operator is invariant under it. In the discrete case, for a vertex-to-vertex mapping $\Pi$ to be isometric, the equivalent statement is the mapping commutes with the Laplace-Beltrami operator for an arbitrary function

$x \in \mathbb{R}^{n_M}$ defined on source mesh $M$. Formally:

$$\Pi\Delta(x) = \Delta(\Pi x) \tag{24}$$

$$\Phi_N C \Phi_M^+ \Delta(x) = \Delta(\Phi_N C \Phi_M^+ x) \tag{25}$$

$$\Phi_N C \underbrace{\Phi_M^+ \Phi_M}_{I_k} \Lambda_M X = \Delta(\Phi_N C \underbrace{\Phi_M^+ \Phi_M}_{I_k} X) \tag{26}$$

$$\Phi_N C \Lambda_M X = \Phi_N \Lambda_N \underbrace{\Phi_N^+ \Phi_N}_{I_k} C X \tag{27}$$

Multiplying both sides on the left by $\Phi_N^+$, we obtain:

$$C\Lambda_M X = \Lambda_N C X. \tag{28}$$

Since this holds for any function $x$, we can minimize $\|\Lambda_N C - C\Lambda_M\|_2^2$ as in equation 1 to obtain a roughly isometric mapping.

### A.5 PERFORMANCE UNDER OCCLUSION

We study the performance of our model under occlusion in two cases.

#### A.5.1 PARTIAL SOURCE AND PARTIAL TARGET

In the first case, both the source and target mesh are partially occluded. As shown in Fig. 11, our model is capable of matching partial meshes reconstructed from RGBD camera captures and finding the correct grasping points.

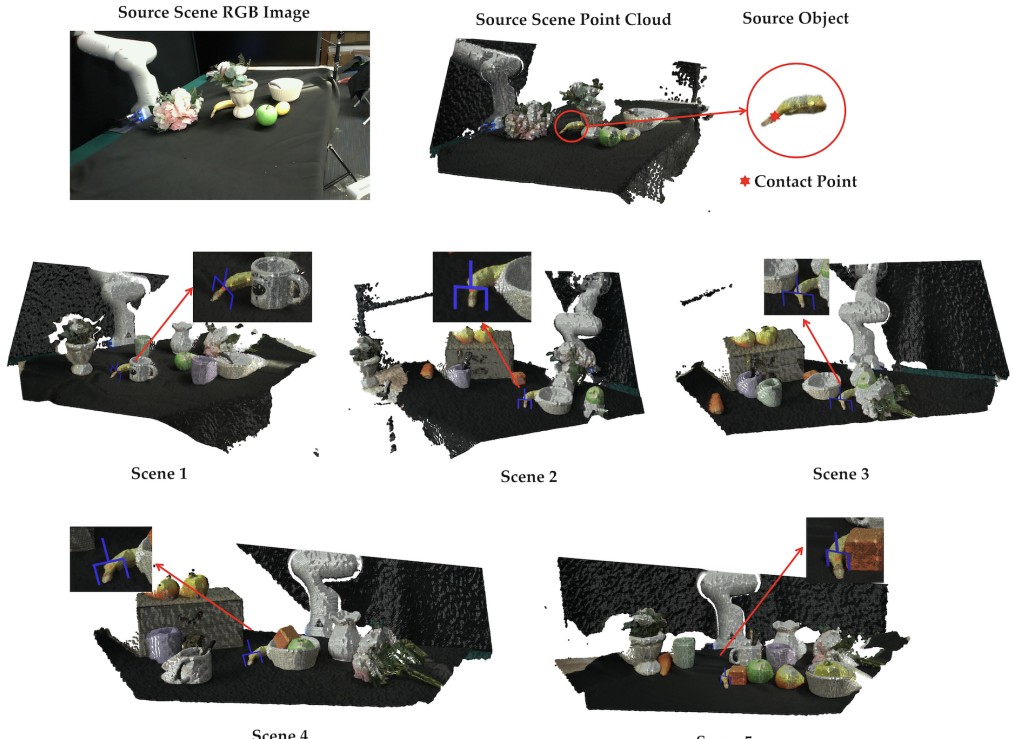

Figure 11: **Robot experiments visualization under occlusion conditions.**

### A.5.2 PARTIAL SOURCE AND FULL TARGET

In the second case, the source mesh is a partial mesh, and the target mesh is a full mesh. We follow the formulation of partial functional correspondence (Rodolà et al., 2017) and jointly optimize a mask $\eta \in \mathbb{R}^{n_N}$ that indicates whether each vertex in the target mesh is matched to the source mesh. In addition to our proposed regularization constraints, we implemented the Mumford-Shah functional and area preservation constraints from Rodolà et al. (2017), in addition to penalizing the entropy of $\eta$ with $-\sum_i^{n_N} \eta_i \log \eta_i$. We showcase qualitative results in Fig. 12 below.

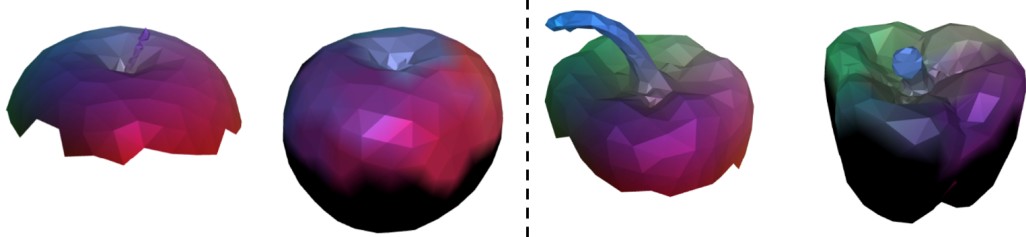

Figure 12: **Correspondence between partial mesh with full mesh.** DenseMatcher is capable of matching a partial mesh to a full mesh by utilizing the partial functional correspondence formulation.

