# OpenReview forum: "DenseMatcher: Learning 3D Semantic Correspondence for Category-Level Manipulation from a Single Demo"
_ICLR.cc/2025/Conference — ICLR 2025 Spotlight_

### Official Review · Reviewer_SE4d · 2024-10-27

**Soundness:** 3
**Presentation:** 3
**Contribution:** 3
**Rating:** 6
**Confidence:** 3

**Summary:**

Summary: This paper introduces DenseCorr3D, a 3D matching dataset featuring colored meshes and dense correspondence annotations. It addresses the limitations of existing datasets that predominantly emphasize geometry. The authors propose DenseMatcher, a model that integrates 2D foundation models with 3D networks to significantly enhance dense correspondence accuracy. The effectiveness of DenseMatcher is demonstrated through applications in robotic manipulation tasks and color transfer experiments.

**Strengths:**

Strengths:

The authors have developed a dataset that is a valuable resource for the research community.

Despite its straightforward pipeline and principles, the proposed DenseMatcher effectively extracts semantic maps that facilitate subsequent tasks.

The introduction of the function map is promising, and the correspondence video demo on the accompanying website is impressive.

**Weaknesses:**

Weaknesses:

The range of tasks and the diversity of object categories provided in the dataset are limited.

Line 853 mentions the total time expenditure without delving into specific details, such as the time required for rendering images, particularly the computation comsumption of the function map.

The paper lacks an ablation study for the DINO and SD components. Previous zero-shot methods shows that the features provided by SD VAE may not be optimal. An ablation analysis for the feature backbone should be included in the experimental tables.

There is no discussion on whether the model incorporates augmentations for the pose of the mesh. Research has shown that semantic features can easily overfit to spatial position-related scenarios. If the input mesh's position changes, the resulting semantic map may become inaccurate. Therefore, it would be beneficial to include experiments that apply random rotations to the mesh as input.

**Questions:**

Additionally, it would be constructive to present examples of failure cases to provide a more comprehensive evaluation.

---

> ### Author Response · Authors · 2024-11-25
>
> **Q1: The range of tasks and the diversity of object categories provided in the dataset are limited.**
>
> Please kindly refer to Common Question in General Response.
>
> **Q2: Line 853 mentions the total time expenditure without delving into specific details, such as the time required for rendering images, particularly the computation comsumption of the function map.**
>
> We thank the reviewer for this valuable suggestion! We have performed inference runtime benchmarking for our model accordingly. We found that for a pair of meshes that are remeshed to ~2000 vertices, computing 2D SD-DINO features for 5 views each consumes ~3.6 seconds, performing DiffusionNet forward pass for each consumes ~0.01 seconds, and computing functional map consumes ~2.2 seconds. For a pair of meshes that are remeshed to ~500 vertices, computing functional map consumes ~0.8 seconds while the other parts remain unchanged. The rendering time for 5 views depends on meshing of the original textured assets and averages to ~0.2 seconds per mesh. We reflected this change in Appendix A.4.3 and Table 5.
>
> **Q3: The paper lacks an ablation study for the DINO and SD components. Previous zero-shot methods shows that the features provided by SD VAE may not be optimal. An ablation analysis for the feature backbone should be included in the experimental tables.**
>
> We acknowledge that SD VAE features lack semantic richness, as discussed in prior works (e.g., Appendix B in DIFT [A]). However, our approach does not utilize VAE features from Stable Diffusion. Instead, we rely on features extracted from the UNet decoder, specifically layers 2, 5, and 8, as outlined in SD-DINO [B] and GeoAware-SC [C], which is shown to be most informative regarding spatial and semantic understanding.
> Our pipeline follows GeoAware-SC for extracting geometry-aware 2D representations and employs their pretrained feature aggregation network, which was trained jointly on both SD and DINOv2 features. Conducting an ablation study for individual feature sets would require retraining the aggregation network for each specific feature set, which is computationally infeasible.
> We direct the reviewer to Table 3 in SD-DINO [B], where a similar ablation study on 2D representations is presented. Their results highlight the significant performance gains achieved by fusing SD and DINO features compared to using either feature set individually.
>
> [A] *Emergent Correspondence from Image Diffusion*. Tang, L., et al. NeurIPS, 2023.
>
> [B] *A Tale of Two Features: Stable Diffusion Complements DINO for Zero-Shot Semantic Correspondence*. Zhang, J., et al. NeurIPS, 2024.
>
> [C] *Telling Left from Right: Identifying Geometry-Aware Semantic Correspondence*. Zhang, J., et al. CVPR, 2024.
>
>
> **Q4: There is no discussion on whether the model incorporates augmentations for the pose of the mesh. Research has shown that semantic features can easily overfit to spatial position-related scenarios. If the input mesh's position changes, the resulting semantic map may become inaccurate. Therefore, it would be beneficial to include experiments that apply random rotations to the mesh as input.**
>
>
> In our original experiments, we randomly rotated the mesh as a part of our training augmentation procedure. We have updated Appendix A.4.2 with this detail.
>
>
> **Q5: Additionally, it would be constructive to present examples of failure cases to provide a more comprehensive evaluation.**
>
> Our unsuccessful cases primarily arise from two sources: one is the inaccurate generation of poses, and the other is the imprecision of the waypoints we provide, which leads to task failure. We have uploaded a case where peeling a banana failed due to the waypoint issue to the linked website（https://densematcher.github.io/）, but this failure is unrelated to the performance of our model. Our model predominantly influences the grasping pose.

---

> > ### Comment · Reviewer_SE4d · 2024-11-26
> >
> > Thanks for the response. I tend to keep the positive rating.

---

> ### Author Response · Authors · 2024-11-27
>
> Thank you for your response. Are there any specific concerns that you feel still place the paper on the borderline for acceptance? We would be happy to discuss and address any remaining issues to further clarify or strengthen our work.
>
> **Additional notes on Q4:**
>
> In Figure 3 of the updated paper, we randomly rotate the mesh before the test procedure to ensure that the model is not reliant on canonical spatial poses. It showcases that our model is robust to random rotations during testing.

---

### Official Review · Reviewer_Jn9K · 2024-11-03

**Soundness:** 2
**Presentation:** 4
**Contribution:** 3
**Rating:** 6
**Confidence:** 3

**Summary:**

This paper studies the problem of dense surface-point matching between objects, where similarity is understood as a user-defined semantic and matches can be between objects of the same, but also different category. The contributed method combines features that encode the visual appearance with features that encode local geometry. The method is evaluated and compared against baselines on a self-created dataset and real-world robotic imitation of human demonstrations.

**Strengths:**

- The authors claim (and I am not aware otherwise, but also not super familiar with this subfield) to contribute the first method for 3D dense correspondences that combined visual appearance and geometric information. This very intuitively makes sense and makes especially the contributed dataset something that can have profound impact on the research on 3D correspondences.
- The method has directly been evaluated in a real-world application of mimicing human demonstrations with a robotic manipulator.
- The paper is very well written (the best in my review batch) and easy to follow.

**Weaknesses:**

- The experimental evaluation is limited to a self-contributed dataset and very few qualitative runs on a robotic application (where it is unclear if the method difference is statistically significant).
- The method design contains a couple of non-straightforward design choices without justifications or experimental evidence to back up these choices [**update from discussion with authors: these points are mostly addressed now**]:
  - Using the XYZ coordinates of the mesh vertices makes the method sensible to random transformations on the input mesh. There is no experiment evaluating whether the model is able to learn invariance over such random coordinate system changes.
  - The choice of negative cosine similarity in $L_\textrm{semantic}$ is quite particular. The authors do not explain why they would choose this over e.g. L1 or L2 distances and also do not ablate this choice.
  - Similarly, for $L_\textrm{preservation}$, the choice of a single linear layer for reconstruction might hinder the encoder network to learn a more useful non-linear function. The more standard choice would probably be to mirror the encoder architecture like in an autoencoder, but this is neither discussed nor evaluated.
- The method requires supervised training with an expensive 3D annotation workflow.

**Questions:**

- Section 4.1: I am not super familiar with the prior work on 3D dense matching, but this optimization formulation seems computationally expensive and as Section 4.4 shows also unstable. Why are other assignment and matching methods not compared as beaseline or ablation? e.g. Hungarian matching or the double-softmax used in [1]?
- line 200: The requirement of textured 3D assets is very limiting. It seems to me the method could also work from an untextured geometry asset and posed images, or am I missing something?
- line 242: Since the negative cosine distance is such an odd choice I suspect the authors were inspired here by related work? In that case it would be important to attribute this here with a reference.
- line 252: "object type and material" is misleading. Neither one of the frozen backbones captures this information, both are self-supervised encoders of visual appearance that might correlate with this information in some cases.
- line 254: What norm is used in the equation for $\mid\mid \cdot \mid\mid$? Why is that one choosen?
- Table 1: Please explain better the different ablation variants. Is "w/o Diffusion Net" directly matching the concatenation of $f_\textrm{multiview}$ and the HKS features? Or is it also using the XYZ features and therefore failing because of coordinate system change?
- Section 6.2.3: I dont't think the comparison to Robo-ABC is entirely fair. It would be good to show both variants, with the full affordance memory and with the reduced form that is currently presented. The proposed method is very expensive in terms of the 3D data it requires, so really it needs to show that this additional information can compete with methods that are only based on cheaper and more abundant image data.
- Section 6.2.4: How is success determined in the experiments? Given the low number of overall trials, what level of statistical significance does the experiment currently have?



[1] Lindenberger, P., Sarlin, P.-E., & Pollefeys, M. (2023). LightGlue: Local Feature Matching at Light Speed. Retrieved from https://openaccess.thecvf.com/content/ICCV2023/html/Lindenberger_LightGlue_Local_Feature_Matching_at_Light_Speed_ICCV_2023_paper.html

---

> ### Author Response · Authors · 2024-11-25
> **Official Comment by Authors (1/3)**
>
> **Q1: The experimental evaluation is limited to a self-contributed dataset and very few qualitative runs on a robotic application (where it is unclear if the method difference is statistically significant).**
>
> For the number of trials in real-world experiments, we follow previous work like Robo-ABC [A] and RAM [B], which is five times.
>
> [A] Robo-ABC: Affordance Generalization Beyond Categories via Semantic Correspondence for Robot Manipulation. Ju, Y., et al. ECCV, 2024.
>
> [B] RAM: Retrieval-Based Affordance Transfer for Generalizable Zero-Shot Robotic Manipulation. Kuang, Y., et al. CoRL, 2024.
>
> **Q2: The method design contains a couple of non-straightforward design choices without justifications or experimental evidence to back up these choices:**
> **Q2(1): Using the XYZ coordinates of the mesh vertices makes the method sensible to random transformations on the input mesh. There is no experiment evaluating whether the model is able to learn invariance over such random coordinate system changes.**
>
> In our original experiments, we randomly rotated the mesh as a part of our training augmentation procedure. In addition to sinusoidal-encoded XYZ coordinates, the 3D refiner network’s input contains heat kernel signature(HKS) descriptors of the scale-normalized mesh, which is invariant to scaling, rotation, and translation. We have updated Appendix A.4.2 to include this detail. Additionally, we showcase in Figure 3 that the model is robust to random rotation during test.
>
> **Q2(2): The choice of negative cosine similarity in $L_\text{semantic}$ is quite particular. The authors do not explain why they would choose this over e.g. L1 or L2 distances and also do not ablate this choice.**
>
> We appreciate the reviewer's thoughtful comments regarding our choice of the negative cosine similarity. We chose this metric because both $||f(v_i) - f(v_j)||$ and $D_\text{semantic}(v_i, v_j)$ are normalized . Under this normalization, optimizing the negative cosine similarity is mathematically equivalent to optimizing the squared $L_2$ distance, while enjoying the benefit of being more interpretable.
>
> To elaborate, for any two normalized vectors $a$ and $b$, their cosine similarity is defined as:
> $$
> \cos(\theta) = \frac{a \cdot b}{||a|| ||b||}.
> $$
> Since $a$ and $b$ are normalized, $||a|| = ||b|| = 1$. The negative cosine similarity becomes:
> $$
> -\cos(\theta) = -a \cdot b.
> $$
> Expanding $||a - b||^2$ for normalized vectors, we have:
> $$
> ||a - b||^2 = ||a||^2 + ||b||^2 - 2a \cdot b = 2 - 2\cos(\theta).
> $$
> Thus, minimizing $-\cos(\theta)$ is equivalent to minimizing the squared $L_2$ distance $||a - b||^2$, up to a constant factor.
>
> In addition, since a maximal cosine similarity implies linear correlation between $||f(v_i) - f(v_j)||$ and $D_\text{semantic}(v_i, v_j)$, we added proof in Appendix A.5.2 to show that optimizing the functional map objective is equivalent to optimizing total $D_\text{semantic}(v_i, v_j)$ beteween matched vertices, and referred to it in Section 4.3.1.
>
> **Q2(3): Similarly, for $L_\text{preservation}$, the choice of a single linear layer for reconstruction might hinder the encoder network to learn a more useful non-linear function. The more standard choice would probably be to mirror the encoder architecture like in an autoencoder, but this is neither discussed nor evaluated.**
>
> We thank the reviewer for this insightful suggestion. We have taken the suggestion and performed experiments with 3 variants of the reconstructor module, where each variant's parameters are optimized together with the model during training. The three variants include (i) a linear layer (ii) a 4-layer MLP, corresponding to the depth of our 3D refiner (iii) a DiffusionNet mirroring the architecture of our 3D refiner. The results are presented below.
>
> | **Variant**         | **AUC ↑** | **Err ↓** |
> |----------------------|:-----------:|:-----------:|
> | **linear (default)**              | **77.5**         | **2.82**         |
> | 4-layer MLP         | 76.6         | 2.94         |
> | mirror              | 53.2         | 5.95         |
>
> We found using reconstructor variant (iii) resulted in poor training loss convergence. We surmise this is because DiffusionNet simulates the heat diffusion process to propagates features on the mesh surface, and attempting to reverse this process forces the diffusion time constant to be near-zero, causing numerical instability.
>
> We observe slightly better performance on benchmark albeit higher training loss when using a linear reconstructor layer compared to a 4-layer MLP. We suspect that this is due to the more powerful MLP reconstructor being prone to overfitting, thus "cheating" the information preservation problem.

---

> ### Author Response · Authors · 2024-11-25
> **Official Comment by Authors (2/3)**
>
> **Q3: The method requires supervised training with an expensive 3D annotation workflow.**
>
> We asked the annotators to time their workflow and found that by annotating sparse keypoints and using heuristics to process them into dense groups, the annotation time is around 10 seconds per mesh, which is on par with labeling image keypoints in 2D. For more complex meshes such as four legged animals which require dense annotation using blender’s vertex brush, the annotation time is around 5 minutes per mesh, which is faster than coarsely-labeled dense annotations [C] on 2D images (~7 minutes per image). We have included the details in Appendix A.2.4 and A.2.5.
>
> In addition, as explained in the paper (Appendix Sec. A.3.2), the only trainable component of our model is the lightweight(~5M parameters) 3D refiner, whose input already contains rich semantic features from frozen 2D foundation models. As a result, our method is highly data-efficient and only requires a few dozen meshes per category to work.
>
> [C] Urban Scene Semantic Segmentation with Low-Cost Coarse Annotation. Das, A., et al. WACV, 2023.
>
> **Q4(1): Section 4.1: I am not super familiar with the prior work on 3D dense matching, but this optimization formulation seems computationally expensive and as Section 4.4 shows also unstable.**
>
> The original motivation of functional map [D] is its computational efficiency, as it decomposes a high-dimensional space of vertices into low-dimensional frequency representations. The theoretical runtime of function map is scales linearly with the number of vertices, and our GPU-accelerated implementation of functional map takes ~0.8 seconds for a pair of meshes with 500 vertices each, and ~2.2 seconds for a pair of meshes with 2000 vertices each. In contrast, other state-of-the-art 3D shape matching methods that solve for globally optimal solutions such as [E] are polynomial time and take >200 seconds for a pair of 2000-vertex meshes. We have added runtime comparisons in Appendix A.4.3 and Table 5.
>
> [D] Functional Maps: A Flexible Representation of Maps Between Shapes. Ovsjanikov, M., et al. ACM Trans. Graph., 2012.
>
> [E] SpiderMatch: 3D Shape Matching with Global Optimality and Geometric Consistency. Roetzer, P., et al. CVPR, 2024.
>
>
> **Q4(2): Why are other assignment and matching methods not compared as beaseline or ablation? e.g. Hungarian matching or the double-softmax used in LightGlue [F]?**
>
>
> We have added a visual comparison of functional map with Hungarian matching and nearest neighbor in Figure 11. Although the two suggested formulations are valuable, we believe they may not be the most suitable for the 3D matching problem. Many 3D objects exhibit circular symmetry, and finding point-to-point correspondences does not necessarily result in a continuous mapping between surfaces.
>
> In addition, following the reviewer's suggestion, we benchmarked the runtime of Hungarian matching on the pairwise vertex feature distance matrix without accounting for spatial consistency, which consumes ~0.01-0.4 seconds for a pair of 500-vertex meshes, and ~0.5-2.5 seconds for a pair 2000-vertex meshes. We have added Table 5 for a straightforward view.
>
> Finally, we have also explored setting up an optimization program that solves for a point-to-point mapping matrix using double softmax while incorporating spatial constraints such as isometry. However, the results did not converge, possibly due to the large search space dimension associated with a point-to-point matrix.
>
>
>
> [F] LightGlue: Local Feature Matching at Light Speed. Lindenberger, P., et al. ICCV, 2023.
>
> **Q5: line 200: The requirement of textured 3D assets is very limiting. It seems to me the method could also work from an untextured geometry asset and posed images, or am I missing something?**
>
> The reviewer is correct that our method would also work from an untextured geometry asset and posed images, as long as they are consistent with each other. We trained our model on textured assets since they are easily sourced from existing datasets. In addition, for future works wishing to scale up this method, state-of-the-art methods in 3D generation such as LRM [G] can already generate high quality textured 3D assets from posed images within seconds, so the limitation is not significant.
>
> [G] LRM: Large Reconstruction Model for Single Image to 3D. Hong, Y., et al. ICLR, 2024.
>
> **Q6: line 242: Since the negative cosine distance is such an odd choice I suspect the authors were inspired here by related work? In that case it would be important to attribute this here with a reference.**
>
> Please kindly refer to Q2(2).

---

> ### Author Response · Authors · 2024-11-25
> **Official Comment by Authors (3/3)**
>
> **Q7: line 252: "object type and material" is misleading. Neither one of the frozen backbones captures this information, both are self-supervised encoders of visual appearance that might correlate with this information in some cases.**
>
> Thank you for your insightful question. We would like to clarify that the Stable Diffusion model is not trained in a self-supervised manner but with billions of text-image pairs, which enables it to distinguish object types and materials.
> In practice, a line of work has shown that material information (e.g., albedo, normal, roughness, and metallic) can be extracted from the Stable Diffusion model [H, I, J, K, L] or the DINOv2 model [K, L], via (LoRA) fine-tuning [H, I, J] or feature probing [K, L]. These studies suggest that material information is present within these visual foundation models.
>
> [H] Intrinsic Image Diffusion for Indoor Single-view Material Estimation, Kocsis et al., CVPR 2024.
>
> [I] RGB↔X: Image Decomposition and Synthesis Using Material-and Lighting-Aware Diffusion Models, Zeng et al., SIGGRAPH 2024.
>
> [J] MaterialFusion: Enhancing Inverse Rendering with Material Diffusion Priors, Litman et al., arXiv 2024.
>
> [K] Probing the 3D Awareness of Visual Foundation Models, El Banani et al., CVPR 2024.
>
> [L] Generative Models: What Do They Know? Do They Know Things? Let's Find Out!, Du et al., arXiv 2023.
>
> **Q8: line 254: What norm is used in the equation for $|| \cdot ||$? Why is that one choosen?**
> Thanks for the feedback. $|| \cdot ||$ denotes L2 norm, following the convention for feature distances. We have updated the paper to use $|| \cdot ||_2$ for clarity.
>
>
> **Q9: Table 1: Please explain better the different ablation variants. Is "w/o Diffusion Net" directly matching the concatenation of and the HKS features? Or is it also using the XYZ features and therefore failing because of coordinate system change?**
>
> The experiment "w/o DiffusionNet" uses only $f_\text{multiview}$ and does not concatenate it with XYZ/HKS features. This was originally mentioned in section 6.4. We have updated Table 1's caption to refer to section 6.4.
>
> **Q10：Section 6.2.3: I dont't think the comparison to Robo-ABC is entirely fair. It would be good to show both variants, with the full affordance memory and with the reduced form that is currently presented. The proposed method is very expensive in terms of the 3D data it requires, so really it needs to show that this additional information can compete with methods that are only based on cheaper and more abundant image data.**
>
> In Table 3, we have added a comparison of the two variants of DenseMatcher and Robo-ABC: one with full memory capabilities and another where Robo-ABC's affordance memory is only allowed to be collected from the corresponding human demos we provide, while keeping Robo-ABC's original retrieval-and-transfer framework intact.
>
> From Table 3, it can be seen that the success rate of Robo-ABC with full memory capabilities is lower than that of Robo-ABC where its affordance memory is only allowed to be collected from the corresponding human demos we provide. Although the amount of data in Robo-ABC's memory is huge, the categories of objects are too different from the object categories used in our experiment. Robo-ABC will retrieve the entire memory for objects it has never seen. In fact, the simplified version of Robo-ABC reduces the error rate of the retrieval process and improves the success rate of the robot experiment. We hope to resolve the reviewer's doubts through the comprehensive experimental results.
>
>
> **Q11: Section 6.2.4: How is success determined in the experiments? Given the low number of overall trials, what level of statistical significance does the experiment currently have?**
>
> Our real-world deployment process is the same as Robo-ABC's, which involves obtaining grasp points on the target object and generating grasp pose at the grasp point using AnyGrasp [M]. The criteria for determining the success of robotic experiments are also the same as Robo-ABC's, which is based on whether the robot grasping is successful.
>
>
> [M] AnyGrasp: Robust and Efficient Grasp Perception in Spatial and Temporal Domains. Fang, H., et al. IEEE Trans. Robotics, 2023.

---

> > ### Comment · Reviewer_Jn9K · 2024-11-28
> > **Thanks for the detailed response**
> >
> > Dear Authors,
> >
> > thanks for your detailed response. It seems to me that most of the points in Weakness 2 are addressed through additional explanations or even experiments, which I am grateful for.
> > My first and main critic, that the experimental evaluation is limited to a self-contributed dataset and very few qualitative runs on a robotic application (where it is unclear if the method difference is statistically significant) still remains.
> > Given that this is however now the only big critic, I am raising my score to borderline accept.

---

> > > ### Author Response · Authors · 2024-12-02
> > >
> > > Thanks for your response and additional feedback.
> > >
> > > Regarding the correspondence experiments, our method specifically targets robotic applications where 3D objects are naturally textured. For this reason, we believe it is more practical to compare 3D matching performance on datasets of textured meshes. To the best of our knowledge, our proposed DenseCorr3D dataset is the only 3D correspondence dataset featuring textured meshes.
> > >
> > >
> > > For the robotic manipulation experiments, we have expanded our evaluation to include three tasks, with the number of trials increased to ten per task. The results, summarized in the table below, demonstrate that as the number of trials increases, our method still significantly outperforms the baselines, highlighting its robustness and stability. Due to time constraints, we will continue to supplement robot experiments after the disccusion phase.
> > >
> > >
> > > |      Method          | **Peeling a banana** | **Pulling out the carrot** | **Flower arrangement** | **Overall** |
> > > |-------------|:----------:|:-----------:|:------------:|:-----------:|
> > > | Robo-ABC（original memory）  | 3/10 | 5/10 | 3/10 | 36.7% |
> > > | Robo-ABC（new memory）  | 7/10 | 6/10 | 3/10 | 53.3% |
> > > | **DenseMatcher (Ours)** | **8/10** | **7/10** | **8/10** | **76.7%** |
> > >
> > >
> > > We hope these additional experiments address your concerns.

---

> > > > ### Comment · Reviewer_Jn9K · 2024-12-02
> > > > **Reviewer response #2**
> > > >
> > > > Thanks authors for the additional response.
> > > >
> > > > I totally get the point that when contributing the first dataset with textured meshes it is hard to compare on another dataset. However, increasing robotic trials from 15 to 30 on a custom robot setup does not increase the rigour or objectivity of the results. Usually in robotics the goal is to combine results from trials with a custom lab setup together with some form of method evaluation on an objective dataset, to make sure that the method is generalizing beyong the setup that the authors are familiar with and prevent overfitting to their lab setup. This 'objectivity check' is discounted a bit when the dataset is contributed in the same paper by the same people.
> > > > Probably a way to provide some more reproducible and objective results would have been to perform robotic simulation experiments in a setup that is already defined by others, such as this task in the SAPIEN simulator that includes textured objects: https://maniskill.readthedocs.io/en/latest/tasks/table_top_gripper/index.html#picksingleycb-v1 I know that this is out of scope of something to do within the discussion period, but its just an example to show that for this work in general, there would have been a way of a more objective evaluation. Therefore I keep my score at borderline accept.

---

> ### Author Response · Authors · 2024-12-04
>
> Thank you for the insightful feedback.
>
> Our focus is on real-world applications, specifically using human hand demonstrations as they are more practical to collect and present greater challenges compared to simulation. These real-world scenarios better reflect whether the method is effective in practical settings. This is why we chose to supplement with additional real-world experiments during the rebuttal.
>
> We agree that testing in simulation could provide additional validation. However, as you mentioned, it is beyond the scope of our work and what we can address within the rebuttal period. We plan to conduct simulation experiments as you suggested, such as those in the SAPIEN simulator, after the discussion phase and will include them in future work.

---

### Official Review · Reviewer_iy2b · 2024-11-04

**Soundness:** 3
**Presentation:** 4
**Contribution:** 3
**Rating:** 8
**Confidence:** 4

**Summary:**

This paper introduces DenseMatcher, an innovative method for computing dense 3D correspondences between objects with similar structures, geared towards applications in robotic manipulation. They propose that *semantic correspondence*—which aligns semantically similar parts across objects—provides more powerful generalization capabilities across categories compared to *shape correspondence*, which mainly focuses on geometry.

To facilitate the training and evaluation, they created *DenseCorr3D*, a new dataset comprising 589 colored object meshes across 23 categories, with dense correspondences organized into semantic group. DenseMatcher utilizes pre-trained 2D foundation models to extract multiview features, which are further refined using DiffusionNet. The enhanced features are then used to establish dense correspondences through a functional map.

They provide comprehensive experiment results to demonstrate DenseMatcher’s effectiveness in 3D dense matching, zero-shot robotic manipulation, and color transfer tasks. DenseMatcher outperformed baseline methods on the DenseCorr3D benchmark and achieved a 76.7% success rate in real-world robotic manipulation, showcasing its robust generalization capabilities.

**Strengths:**

- Integration of 2D and 3D: DenseMatcher effectively combines 2D foundation models, like SD-DINO, for multiview feature extraction with DiffusionNet to refine features with geometry. This fusion enhances semantic understanding and generalizability in 3D correspondence.

- New 3D matching dataset: The authors introduce DenseCorr3D, the first dataset with colored meshes and dense correspondences, featuring 589 textured meshes across 23 categories. It advances research by supporting methods that account for both appearance and geometry.

- Enhanced functional map for accuracy: A novel regularization scheme promotes sparsity in DenseMatcher’s functional map, achieving a 43.5% accuracy improvement over baselines.

- The paper is well-written and easy to understand. The experiment results are comprehensive and promising.

**Weaknesses:**

- Limited analysis on varying topologies: While they analyze that previous methods struggle with different topologies, they do not deeply explore DenseMatcher's robustness on diverse object structures.

- Limitation to severe occlusion: The paper does not address how DenseMatcher handles significant occlusion. Since it relies on multiview feature extraction and functional maps, both susceptible to occlusion, further analysis of this limitation would strengthen the evaluation.

**Questions:**

- Performance on Varying Topologies: How does DenseMatcher perform with objects of varying topologies? Are there specific object structures or topological variations where its performance significantly degrades?

- Handling Severe Occlusion: Is DenseMatcher able to be adapted or extended to handle severe occlusion more effectively? What potential modifications could mitigate its reliance on multiview feature extraction and functional maps in such cases?

- More Benchmark Validation: Are there any benchmarks or experiments that could further validate DenseMatcher’s robustness against topological diversity and occlusion? How might these additional evaluations impact its overall effectiveness and applicability in real-world scenarios?

---

> ### Author Response · Authors · 2024-11-25
>
> **Q1: Performance on Varying Topologies: How does DenseMatcher perform with objects of varying topologies? Are there specific object structures or topological variations where its performance significantly degrades?**
>
> The daily object subset of our dataset contains chairs that each have 4 legs and a backrest made of planks with holes in between, animals with 4 legs, and cars that are empty inside and have holes in windows. We have added Figure 3 and Table 6 (as below) to provided more visualizations and quantitative test results for those categories.
>
> |               | **Chairs** | **Animals** | **Broccoli** | **Shampoo** |
> |:-------------:|:----------:|:-----------:|:------------:|:-----------:|
> | URSSM  | 4.71 | 6.75 | 7.55 | 4.93 |
> | **DenseMatcher (Ours)** | **3.51** | **3.21** | **3.06** | **3.15** |
>
> *Table: 3D correspondence performance (Error $\downarrow$) on categories with complex topologies.*
>
>
> **Q2: Handling Severe Occlusion: Is DenseMatcher able to be adapted or extended to handle severe occlusion more effectively? What potential modifications could mitigate its reliance on multiview feature extraction and functional maps in such cases?**
>
> In the original training procedure, with a 50% probability, we sliced the mesh along random directions and removed half of it, in order to simulate occlusion. We have added this detail in our appendix along with other training augmentations. During inference, without any extra modification, our model is capable of establishing correspondences from one partial mesh to another partial mesh. As a result, in our real-world robot experiments, we did not use a multi-perspective camera setup but relied solely on a single L515 camera, which only captures the camera-facing part of the mesh. Following the review, we ran additional robotic experiments by intentionally putting obstacles that occluded the object. We have updated Figure 12 in the appendix to demonstrate that even under severe occlusions, our model can achieve successful grasps. In Figure 12, the red dots represent contact points, and the blue poses represent the poses generated at these contact points. Further, to handle cases where we need to match whole objects to partial objects, we have re-implemented a version of partial functional map [A], which co-optimizes a mask on the full mesh with the functional map itself. We have provided more explanations and visualizations in Section A.6 and Figure 13.
>
> [A] *Partial Functional Correspondence* Rodolà, E., et al. Computer Graphics Forum, 2017.
>
> **Q3: More Benchmark Validation: Are there any benchmarks or experiments that could further validate DenseMatcher’s robustness against topological diversity and occlusion? How might these additional evaluations impact its overall effectiveness and applicability in real-world scenarios?**
>
> Please kindly refer to Common Question in General Response.

---

> > ### Comment · Reviewer_iy2b · 2024-11-25
> >
> > Thank you to the authors for the detailed rebuttal and experiments! I would like to maintain my rating, as I believe this is a strong paper overall, with no significant concerns from my perspective. I am happy to recommend it for acceptance.

---

> > > ### Author Response · Authors · 2024-11-27
> > >
> > > Thank you for your follow-up and recognition of the updates. We appreciate your constructive feedback!

---

### Official Review · Reviewer_w2Z1 · 2024-11-04

**Soundness:** 4
**Presentation:** 4
**Contribution:** 4
**Rating:** 10
**Confidence:** 5

**Summary:**

This paper proposes a framework and dataset for category-level object 3D dense matching. The DenseMatcher utilizes a 2D foundation model with 3D network refinement to reach generalization and 3D understanding. The author conducts robotic manipulation and zero-shot color mapping to validate the findings.

**Strengths:**

1. This idea is novel and underexplored in relevant areas, especially in robotic manipulation learning. Instead of simply augmenting the data with numerous demos, this paper can address sample efficiency by embedding semantic information.

2. This paper's writing style is straightforward, and it is easy to catch the main topic.

3. Utilizing the existing 2D network (DINO in this paper) with 3D networks is a simple but promising approach.

4. Experiments can thoroughly reflect the model's ability. In robotic manipulation tasks, it covered pick-and-place, long-horizon, and dual arm.

**Weaknesses:**

1. The statements of regularization terms in the methodology part are unclear and may cause ambiguity.

2. Some experiment details, like the description for each task, can be placed in the appendix and give a more precise visualization. The images in the robotic manipulation task are too undersized.

**Questions:**

1. In Sec 4.1 Preliminary, Functional Map, please give a detailed justification about how to regularize the term C as isometric in your context.

2. In the appendix, please provide a detailed explanation, with proofs, showing how previous constraint terms ensure that the output is minimized in the semantic distance function.

---

> ### Author Response · Authors · 2024-11-25
>
> **Q1: The statements of regularization terms in the methodology part are unclear and may cause ambiguity. In Sec 4.1 Preliminary, Functional Map, please give a detailed justification about how to regularize the term C as isometric in your context.**
>
> We appreciate the insightful feedback. We added derivations for isometric regularization of the term C in Appendix A.5.3 and refered to it in Sec 4.1. This addition aims to enhance clarity and address any potential ambiguities in our methodology. Thank you for highlighting this concern.
>
> **Q2: In the appendix, please provide a detailed explanation, with proofs, showing how previous constraint terms ensure that the output is minimized in the semantic distance function.**
>
> Thank you for your valuable input. In Appendix A.5.2, we added detailed derivations for the previous constraint terms. In addition, we showed with proofs that our proposed semantic distance function is minimized under the functional map framework.
>
> **Q3: Some experiment details, like the description for each task, can be placed in the appendix and give a more precise visualization. The images in the robotic manipulation task are too undersized.**
>
> Following your suggestion, we have reorganized the structure of the paper, placed the description of  tasks in the appendix, and re-uploaded Figure 8 to provide a clearer view of the robotic experiments.

---

> > ### Comment · Reviewer_w2Z1 · 2024-11-27
> >
> > Thank you to the authors for providing a detailed explanation that addressed all my concerns. I have updated my score to a strong acceptance and will advocate for acceptance in future discussions.

---

> > > ### Author Response · Authors · 2024-11-27
> > >
> > > Thank you very much for your kind support and positive feedback! We are truly grateful for your thorough review and valuable suggestions, which greatly helped us improve our work!

---

### Author Response · Authors · 2024-11-25

# General Response

We thank the reviewers for their insightful comments and recognition of our work's strengths:

- **Novel Contribution:** Reviewers highlighted our work as "novel and underexplored," particularly in robotic manipulation learning (w2Z1), and the design of functional map is also noted as a "novel regularization scheme" (iy2b).
- **Effective Integration of 2D and 3D Models:** Our approach of combining 2D foundation models with 3D networks was praised as enhancing "semantic understanding and generalizability" (iy2b) and as a "simple but promising approach" (w2Z1).
- **Valuable Dataset Introduction:** The dataset we developed was recognized as a "valuable resource for the research community" (SE4d) and for "advancing research by supporting methods that account for both appearance and geometry" (iy2b), "profound impact on the research on 3D correspondences" (Jn9K).
- **Clear Presentation:** The paper was commended for being "straightforward and easy to catch the main topic" (w2Z1), "very well written and easy to follow" (Jn9K), and "well-written and easy to understand" (iy2b).
- **Strong Experimental Validation:** Reviewers appreciated that our experiments "thoroughly reflect the model's ability," covering various tasks (w2Z1), and demonstrating effectiveness in "zero-shot robotic manipulation" (iy2b). The real-world application and the "impressive" video demo were also noted (Jn9K, SE4d).

------

We have uploaded the detailed explanations requested by the reviewers, along with additional experiments, including the results of real-world robot experiments. We hope these experiments address the reviewers' concerns. All revisions are highlighted in **red** in the updated version.
- **Model performance for more diverse topologies(iy2b-Q1; Figure 3, Figure 4, Table 6):** Added figures to show qualitative results and training data examples. We also provided numerical results in the Appendix Table 6.
- **Proofs and derivations of constraints for functional map(w2Z1-Q1, Q2; Section 4.1, Appendix A.5):** provided highly detailed derivations in appendix A.5 and referenced it in Section 4.1.
- **Justification of cosine similarity loss(Jn9K-Q2(2), Q6; Section 4.3.1, Appendix A.5.2):** proved in A.5.2 that after minimizing the cosine similarity objective described in 4.3.1, solving functional map is equivalent to minimizing total $D_\text{semantic}$ between matched vertices.
- **Explanation of ablation variants(Jn9K-Q9; Table 1, Section 6.4):** Referenced Section 6.4 in Table 1's caption.
- **Comparison with Robo-ABC:(Jn9K-Q10; Secion 6.2.2 and Table 3):** Supplemented with experimental results of two variants of Robo-ABC.
- **Reorganize the paper structure(w2Z1-Q3; Line 789 and Figure 8):** Placed the task description section in the appendix and re-uploaded a clearer Figure 8.
- **Comparison with Hungarian matching(Jn9K-Q4; Secion 6.5 and Figure 11):** added a paragraph to explain the advantage of functional map and added a figure to compare results with Hungarian matching.
- **Diversity of objects(iy2b-Q3, SE4d-Q1; Appendix A.2.1 and Table 4):** added to our dataset and updated the table to include more details about the diversity of our dataset.
- **Annotation Cost(Jn9K-Q3; Appendix A.2.4 and A.2.5):** added details about the annotation process and time consumption.
- **Training details(Jn9K-Q2, SE4d-Q4; Appendix A.4.2):** elaborated training procedure and augmentations details.
- **Runtime Analysis(SE4d-Q2; Appendix A.4.3 and Table 5):** provided detailed runtime analysis of our method and baselines
- **Handling Severe Occlusion(iy2b-Q2&Q3; A.6, Figure 12, and Fig13):** Supplemented visualizations of robotic and matching experiments under different severe occlusion conditions.
- **Examples of failure cases(SE4d-Q5；Our website):** Uploaded a failed robotics experiment video.
- **Update L2 norm notation(Jn9K-Q8)**: changed the notation of L2 norm from $|| \cdot ||$ to $|| \cdot ||_2$

------

Below, we address a common question raised by multiple reviewers, while detailed responses to specific reviewers are provided in separate posts.

**Common Question (iy2b-Q3, SE4d-Q1): Diversity of objects**

Appendex A.2.1 and Table 4 show that our dataset contains daily object categories spanning nearly two dozens of fruits & vegetables, vehicles, animals, backpacks, tools, and toiletries. Some categories actually contain more sub-types that were not listed(for example, “animals” contain 9 distinct species such as elephant, giraffe, cat, deer, dinosaur, etc). In addition, we have added a chairs category. We haved updated the category list in Appendix A.2.1 to include object subtypes.

---

### Meta-Review · Area_Chair_eu6r · 2024-12-20

**Metareview:**

This paper introduces DenseMatcher to compute dense surface-point matching between objects, aiming at robotic manipulation applications. This proposed method significantly enhances semantic understanding and generalizability in dense 3D correspondence tasks, thereby improving the downstream manipulation performance. This paper is well written, supported by novel ideas, adequate experiments, and tangible contributions (dataset, method). Therefore, I recommend accepting this paper. I encourage authors to incorporate reviewers' comments in the final revision.

**Additional Comments On Reviewer Discussion:**

After rebuttal, reviewers unanimously decide to accept this paper. There is no significant further concern raised in the rebuttal process.

---

### Decision · Program_Chairs · 2025-01-22

Accept (Spotlight)